# Divergent sequences of tetraspanins enable plants to specifically recognize microbe-derived extracellular vesicles

Jinyi Zhu [1], Qian Qiao[1], Yujing Sun[1], Yuanpeng Xu [1], Haidong Shu [1], Zhichao Zhang[1], Fan Liu[1], Haonan Wang[1], Wenwu Ye [1,2,3], Suomeng Dong [1,2,3], Yan Wang [1,2,3], Zhenchuan Ma[1,2,3] & Yuanchao Wang [1,2,3] ✉

Extracellular vesicles (EVs) are important for cell-to-cell communication in animals. EVs also play important roles in plant–microbe interactions, but the underlying mechanisms remain elusive. Here, proteomic analyses of EVs from the soybean (*Glycine max*) root rot pathogen *Phytophthora sojae* identify the tetraspanin family proteins PsTET1 and PsTET3, which are recognized by *Nicotiana benthamiana* to trigger plant immune responses. Both proteins are required for the full virulence of *P. sojae*. The large extracellular loop (EC2) of PsTET3 is the key region recognized by *N. benthamiana* and soybean cells in a plant receptor-like kinase NbSERK3a/b dependent manner. TET proteins from oomycete and fungal plant pathogens are recognized by *N. benthamiana* thus inducing immune responses, whereas plant-derived TET proteins are not due to the sequence divergence of sixteen amino acids at the C-terminal of EC2. This feature allows plants to distinguish self and non-self EVs to trigger active defense responses against pathogenic eukaryotes.

Extracellular vesicles (EVs) are membrane structures enclosed by a phospholipid bilayer[1–3]. EVs are emerging as important players in plant–microbe interactions[1,4,5]. Microbial infection stimulates plants to secrete increased levels of EVs. Once plant cells detect microbial infection, late endosomes are redirected to transport defense substances to the infection site to hinder further microbial spread[6]. The EVs secreted by the model plant Arabidopsis (*Arabidopsis thaliana*) contain many stress response and disease resistance-related proteins[7]. In addition, substances identified in plant EVs can directly inhibit pathogen growth[1,8,9]. Plant EVs also mediate trans-kingdom transport at plant–microbe interfaces[10,11], and pathogen EVs function as virulence factors[12,13]. Oomycete pathogens are thought to use EVs to secrete RxLR effectors as an unconventional secretion pathway[14,15].

Tetraspanin (TET) family proteins are widely distributed in eukaryotes. In mammals, the tetraspanins CD9, CD37, CD63, CD81 and CD82 are located on the membranes of EVs and are often used as EV marker proteins[16,17]. Similarly, in Arabidopsis, the EV-localized proteins AtTET8 and AtTET9 have been used as EV markers[10]. TET proteins contain four transmembrane regions, with both their N terminus and C terminus located inside the membrane and two loops exposed outside the membrane[18]. Four TET families have been identified in filamentous fungi[19]: Pls1, Tsp2, Tsp3, and the tetraspanin-like family Tpl1. None of these fungal tetraspanins have been identified yet in fungal EVs, but the more widely conserved Pls1 family is a plausible candidate for this role.

As a countermeasure, microbial-secreted EVs can be detected by plant immune systems to elicit defense responses. For example, EVs secreted by *F. oxysporum* f. sp. *vasinfectum* induced cell death in plant leaves[20]. Outer membrane vesicles (OMVs) released by various bacteria act as elicitors to induce plant immune responses, which requires the co-receptor BRI1-ASSOCIATED KINASE 1 (BAK1)[21,22]. OMVs were recently shown to enhance plant immunity in a remorin- and sterol-dependent manner[23]. Nevertheless, the components of microbial EVs that induce plant immunity remain unknown.

[1]Department of Plant Pathology, Nanjing Agricultural University, 210095 Nanjing, China. [2]The Key Laboratory of Integrated Management of Crop Diseases and Pests (Ministry of Education), 210095 Nanjing, China. [3]The Key Laboratory of Plant Immunity, Nanjing Agricultural University, 210095 Nanjing, China. ✉ e-mail: wangyc@njau.edu.cn

In this study, we used ultracentrifugation to isolate EVs from culture filtrates of *P. sojae*, an oomycete pathogen of soybean (*Glycine max*). We verified the EVs by electron microscopy and nanoparticle tracking analysis. Proteomic analysis of *P. sojae* EVs identified PsTET1 and PsTET3. These TET proteins are redundantly required for the full virulence of *P. sojae* and trigger plant defense responses in a BAK1-dependent manner. Furthermore, many TET proteins from oomycetes and fungi elicited plant immune responses, whereas plant-derived TET proteins uniformly failed to do so. Due to the divergent amino acid sequences at the C terminus of the EC2 domain, plants are unable to respond to tetraspanins produced by plants. In addition, purified *P. sojae* EVs triggered immune responses in plant leaves, whereas EVs from *N. benthamiana* and soybean did not. Knocking out both *PsTET1* and *PsTET3* significantly reduced the generation of reactive oxygen species (ROS) induced by *P. sojae* EVs. Therefore, differences between EV tetraspanins appear to be the main mechanism by which plant innate immune systems distinguish between self and non-self EVs, triggering active defense responses against pathogens.

## Results

### PsTET1 and PsTET3 induce cell death in a NbSERK3a/b-dependent manner in *N. benthamiana*

We isolated and purified EVs from *P. sojae* culture filtrates by ultracentrifugation and sucrose gradient centrifugation to investigate their functions. We observed vesicle-like structures using negative staining and transmission electron microscopy (Supplementary Fig. 1a–c). We estimated the sizes of the purified EVs by NTA. Most EVs were in the 60- to 200-nm range, with the main peak around 112 nm. The liquid culture medium itself contained very low concentrations of EVs (Supplementary Fig. 1d).

To determine whether EV proteins can trigger plant responses, we analyzed proteins from purified *P. sojae* EVs by liquid chromatography/tandem mass spectrometry (LC/MS-MS). We identified 468 EV proteins with more than three unique peptides detected in each of the three repeats (Supplementary Data 1). Gene Ontology (GO) term enrichment analysis showed that several proteins in the EV proteome are predicted to have catalytic activity or antioxidant activity (Fig. 1a). Compared to the whole proteome of *P. sojae*, we identified EV proteins enriched in GO terms in the molecular function, cellular component, and biological process categories (Supplementary Fig. 2a). Of the 468 EV proteins identified, 304 lacked a signal peptide (Supplementary Fig. 2b). In addition, we identified two RxLR effectors in the EV proteome (Supplementary Data 1). We reasoned that proteins exposed to the outside of the EV membrane might be more easily recognized by plants than proteins inside the EVs. We therefore focused on transmembrane proteins from EVs. Of the 468 proteins, 95 contained predicted transmembrane regions.

According to transcriptome data from *P. sojae* at different stages of infection, 28 of the 95 genes encoding EV-located proteins with predicted transmembrane regions were expressed during early infection (Fig. 1b and Supplementary Data 2). We cloned these genes and transiently expressed them individually in *N. benthamiana* leaves to assay their ability to induce cell death. The expression of two candidates (Ps_136802 [named PsTET1] and Ps_155746 [named PsTET3]) belonging to the tetraspanin family clearly induced cell death when expressed in *N. benthamiana* leaves (Fig. 1c, d). Ion leakage tests confirmed that both PsTET1 and PsTET3 induce cell death when transiently expressed in *N. benthamiana* (Supplementary Fig. 3). The levels of all proteins were confirmed by immunoblot analysis with anti-GFP antibodies (Supplementary Fig. 4). These results indicate that PsTETs from *P. sojae* EVs can induce cell death in *N. benthamiana*.

BAK1 is an important plant cell membrane-localized co-receptor that is essential for plant responses to many different pathogen-associated molecular patterns (PAMPs). To investigate whether the plant responses induced by PsTETs depend on BAK1 (NbSERK3a/b in *N. benthamiana*)[24], we expressed *PsTET1* and *PsTET3* in *NbSERK3a/b*-silenced *N. benthamiana* leaves. The *P. sojae* elicitor PsXEG1 (which requires BAK1) was used as a positive control, whereas NPP1 (necrosis-inducing *Phytophthora* protein, which does not require BAK1) was used as a negative control[25]. PsTET1 and PsTET3 did not induce cell death in *NbSERK3a/b*-silenced *N. benthamiana* plants, but they still induced cell death in TRV: *GFP* plants expressing a control construct inducing silencing of the green fluorescent protein (*GFP*) gene (Fig. 2a). The silencing efficiency of *NbSERK3a/b* mediated by virus-induced gene silencing reached 80% (Fig. 2b). The proteins were well expressed in both *NbSERK3a/b*-silenced plants and control plants (Fig. 2c). In addition, PsTET1 and PsTET3 colocalized with the plant EV marker protein AtTET8 inside plant cells and with AtTET8 in the secreted EVs (Fig. 2d and Supplementary Fig. 5a). We also confirmed the targeting of *N. benthamiana* EVs by PsTET1 and PsTET3 via immunoblotting (Fig. 2e). The EVs were isolated from *N. benthamiana* before cell death (Supplementary Fig. 5b).

### PsTET1 and PsTET3 are redundantly required for the full virulence of *P. sojae*

*PsTET1* and *PsTET3* expression was upregulated during the early infection of *P. sojae* (Fig. 3a), suggesting that the two encoded proteins may contribute to *P. sojae* infection. To examine the functions of these two TET proteins, we knocked out *PsTET1* and *PsTET3* in *P. sojae* strain P6497 using clustered regularly interspaced short palindromic repeat (CRISPR)/CRISPR-associated nuclease 9 (Cas9)-mediated gene editing[26,27] (Supplementary Fig. 6a–c). Knockout of *PsTET1* or *PsTET3* individually had no significant effect on the growth or virulence of *P. sojae* (Supplementary Fig. 7a–d). Therefore, we knocked out *PsTET1* and *PsTET3* simultaneously in P6497 and obtained two double knockout mutants, named dko-3 and dko-37. The two double knockout mutants grew normally on V8 agar medium, and their growth rates were not significantly different from that of the wild type (Supplementary Fig. 7a–c). However, the virulence of the double knockout mutants was significantly lower than that of the wild type (Fig. 3b, c). To investigate how these two TET proteins contribute to *P. sojae* virulence, we measured the levels of EVs released by the knockout mutants using NTA. The growth of the mutants in a liquid medium resembled that of the wild type (Supplementary Fig. 8a, b). However, the levels of EVs released by the double knockout mutants significantly decreased compared to the wild type (Fig. 3d, Supplementary Fig. 8c), whereas the levels of EVs released by the single knockout mutants were similar to the wild type (Supplementary Fig. 8c, d).

To determine the localization of PsTET3, we generated a construct encoding a fusion of eGFP fused to the N terminus of PsTET3 and used it to transform *P. sojae* strain P6497. We obtained a single transformant that stably expressed e*GFP-PsTET3* (Supplementary Fig. 9a–c). We cultured the transformant in liquid synthetic medium and extracted EVs from the culture filtrate. We detected eGFP-PsTET3 in the P100 fraction, but not the S100 fraction, by immunoblotting using anti-GFP antibodies (Fig. 3e). To ascertain that eGFP-PsTET3 was protected by the membranes of EVs, we treated isolated EVs with the protease trypsin for 30 min or with pretreatment with Triton X-100 to destroy the membrane structure. Following trypsin treatment, we observed the degradation of eGFP-PsTET3, leaving an eGFP-sized band (Fig. 3f). These results suggest that the external region of the eGFP-PsTET3 fusion protein was degraded, leaving the internally localized eGFP domain plus the PsTET3 internal domain protected from trypsin digest by membrane structures. Furthermore, the density of eGFP-PsTET3-enriched EVs was 1.141 g/ml, as revealed by sucrose gradient centrifugation (Fig. 3g), which is similar to the density reported for AtTET8-positive EVs[28].

To examine the release of EVs by *P. sojae* during soybean infection, we infected the hypocotyls of etiolated soybean seedlings with zoospores of the transformant expressing *eGFP-PsTET3* and observed

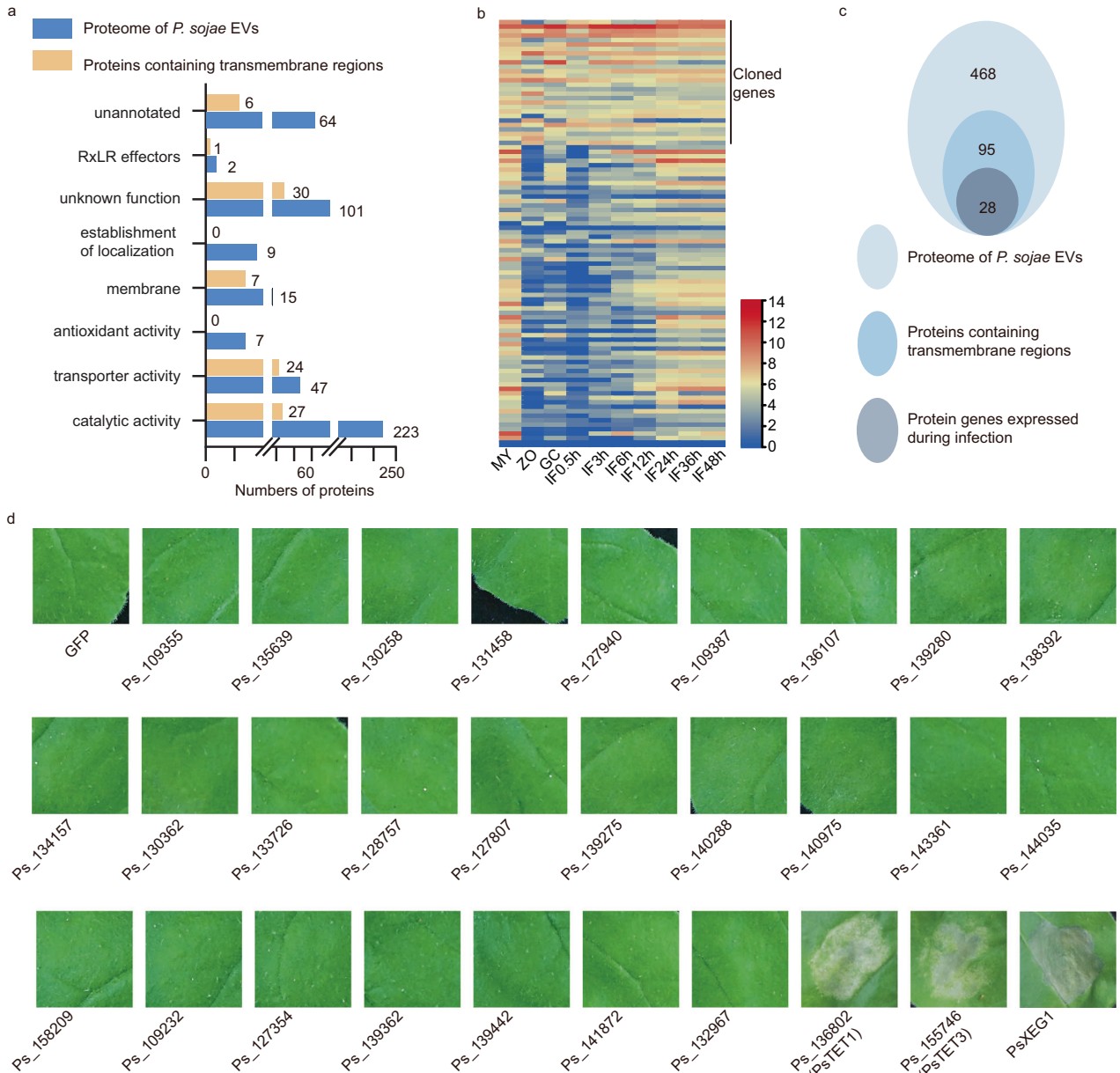

**Fig. 1 | Tetraspanin family proteins can induce cell death in *N. benthamiana*.**
**a** GO functional analysis of proteins and proteins containing transmembrane region in EVs of *P. sojae*. **b** Transcript levels of 95 genes at different stages (My, non-sporulating mycelium grown in V8 broth; ZO, zoospores; GC, germination of cysts; IF3 = 3 h post-inoculation). **c** Classification of proteins detected in EVs of *P. sojae*.

**d** Cell death assays of *P. sojae* EV proteins expressed in *N. benthamiana* leaves. PsXEG1 is a cell death inducing control rather than the candidate of EV protein. *N. benthamiana* leaves infiltrated with indicated constructs were photographed 3 days after agro- infiltration and representative leaves are shown. Experiments were repeated three times with similar results.

epidermal cells under a confocal microscope. We detected numerous tiny green fluorescent spots adjacent to the *P. sojae* infection sites, but not at infection sites of the control strain accumulating cytoplasmic eGFP (Fig. 3h). Some green spots appeared to be located in the cytoplasm of soybean cells. To determine whether the green spots were eGFP-PsTET3-labeled EVs, we observed EVs isolated from the culture medium of *eGFP-PsTET3*-expressing transformants via confocal microscopy. We observed numerous green spots on the surface of the slide (Fig. 3i). When we stained eGFP-PsTET3-labeled EVs with FM4-64, the green spots also showed red fluorescence (Fig. 3i), further confirming that the green spots we observed were indeed EVs.

NTA showed that the *eGFP-PsTET3*-expressing transformant released more EVs than wild-type *P. sojae* (Supplementary Fig. 9d, e). However, the virulence of the transformant was significantly reduced

(Supplementary Fig. 9f, g). These results indicate that high levels of eGFP-PsTET3-labeled EVs hinder infection by *P. sojae*.

## EC2 is necessary and sufficient for PsTET3 recognition by *N. benthamiana* and soybean

Like other tetraspanins[29–31], PsTET3 is predicted to have four transmembrane regions, with it N and C termini located inside the membrane and two loops located outside the membrane (Supplementary Fig. 10). To determine whether the two external loops of PsTET3 could be recognized by *N. benthamiana*, we constructed the variant protein T3M1 by replacing the small extracellular loop (EC1) with a flag tag and the T3M2 variant by replacing the large extracellular loop (EC2) with an eGFP tag (Fig. 4a, b). When constructs encoding these variants were expressed in *N. benthamiana*, T3M2 failed to trigger cell death, whereas T3M1 still induced cell death, indicating that EC2 is essential

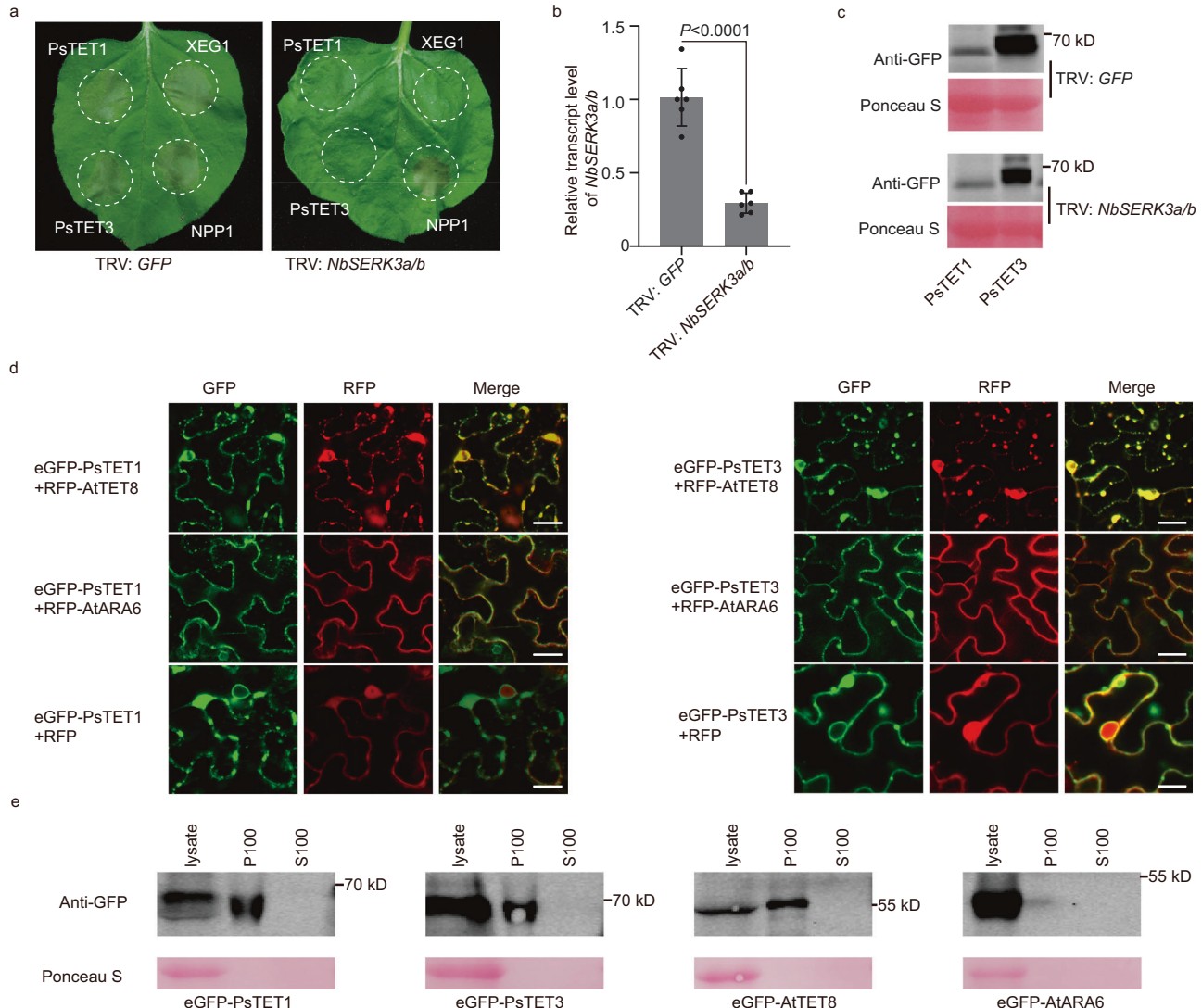

**Fig. 2 | Responses to PsTET1 and PsTET3 require NbSERK3a/b. a** Representative leaves showing cell death induced by expression of PsTET3, XEG1 or NPP1 in *NbSERK3a/b*-silenced (TRV: *NbSERK3a/b*) or control (TRV: *GFP*) *N. benthamiana* leaves. Leaves (*n* = 9) were photographed three days after agro-infiltration. **b** RT-qPCR quantification of transcript levels of *NbSERK3a/b* in VIGS-silenced *N. benthamiana* leaves, normalized to those in TRV: *GFP* lines, using the *NbEF1α* gene as an internal reference. Data are presented as mean values (±SD). Statistical analyses were performed using Two-tailed Student's *t* test. **c** Immunoblot analysis of PsTET1 and PsTET3 fused with an eGFP tag transiently expressed in TRV: *GFP* or TRV: *NbSERK3a/b N. benthamiana* leaves for 2 days after agro-infiltration. Ponceau S-stained Rubisco protein is shown as a total protein loading control. **d** Confocal

microscopy images of eGFP-PsTET1/eGFP-PsTET3 and plant extracellular vesicles marker protein RFP-AtTET8 and multivesicular bodies (MVB) marker protein RFP-AtARA6 and RFP as control in *N. benthamiana* leaves. eGFP-PsTET1 and eGFP-PsTET3 were colocalized with AtTET8 and were partially colocalized with AtARA6. Scale bars, 20 μm. **e** PsTET1 and PsTET3 are localized to EVs when expressed in *N. benthamiana*. EVs of *N. benthamiana* are isolated from apoplastic fluids of leaves, before the development of cell death. AtTET8 is a positive control and AtARA6 is a negative control. Ponceau S-stained Rubisco protein is shown as a total protein loading control. All experiments were repeated three times with similar results. Source data are provided as a Source Data file.

for the recognition of PsTET3 (Fig. 4a, b). To further investigate the role of EC2 in inducing cell death, we constructed two additional variant constructs, T3M3 and T3M4, encoding protein variants with partial truncations of EC2 (Supplementary Data 3). PsTET3-induced cell death was abolished when the C terminus of EC2 was removed (T3M3), but not when its N terminus was removed (T3M4) (Fig. 4a, b). All four variant proteins accumulated at adequate levels in *N. benthamiana* leaves (Supplementary Fig. 11a). We obtained similar results when we analyzed PsTET1 (Supplementary Fig. 11b, c). The localization of the PsTET3 variants was unchanged compared to native PsTET3 (Supplementary Fig. 12), and PsTET3M2 and PsTET3M3 still colocalized with AtTET8 and were targeted to the EVs of *N. benthamiana* (Supplementary Fig. 13a–c). To determine whether EC2 could trigger a plant immune response, we fused EC2 from PsTET3 with a His tag at its

C terminus and expressed the construct encoding this fusion protein in the yeast *Pichia pastoris* (Supplementary Fig. 14a). Recombinant purified EC2-His protein induced a ROS burst in *N. benthamiana* leaf discs in a dose-dependent manner (Supplementary Fig. 14b). Heat treatment did not influence the EC2-triggered ROS burst (Supplementary Fig. 14c).

When treated with recombinant purified EC2-His, the ROS burst induced in leaf discs from *NbSERK3a/b*-silenced *N. benthamiana* leaves was significantly reduced compared to that of the control (Fig. 4c, d). Three PAMP-triggered immunity (PTI) marker genes[32], *NbCYP71D20*, *NbPTI5*, and *NbWRKY7*, were upregulated in *N. benthamiana* leaves infiltrated with 1 μM purified EC2-His. This response diminished in *NbSERK3a/b*-silenced *N. benthamiana* leaves (Fig. 4e and Supplementary Fig. 14d, e). Furthermore, following treatment with 1 μM

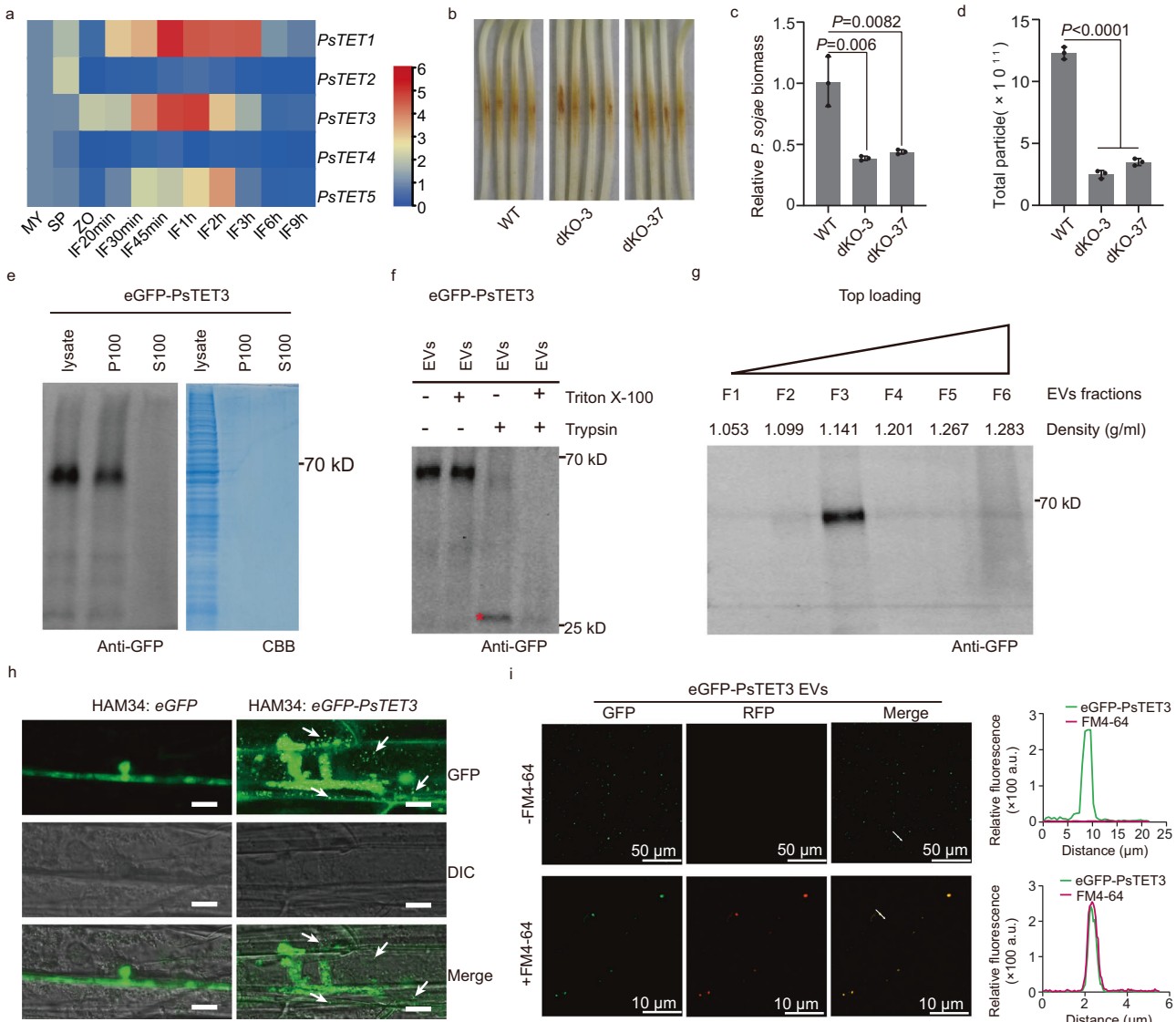

**Fig. 3 | EV-localized PsTET1 and PsTET3 proteins are required for the full virulence of _P. sojae._ a** Transcript levels of _PsTET_ genes at different stages (My, non-sporulating mycelium grown in V8 broth; SP, sporangium; ZO, zoospores; IF20min = 20 min post-inoculation). Transcript levels measured by RT-qPCR were normalized to levels in mycelium using the _PsActin_ gene as an internal reference. The data are presented as fold changes in a heat map. **b, c** _P. sojae PsTET1_ and _PsTET3_ double knockout mutants exhibit reduced virulence. **b** Disease symptoms photographed at 2 days post-inoculation. **c** Relative _Phytophthora_ biomass in inoculated etiolated hypocotyls measured as the ratio between the amounts of _P. sojae_ DNA and soybean DNA assayed at 2 dpi by qPCR; levels in P6497-inoculated soybean were set to 1.0. Data are presented as mean values (±SD). Statistical analyses were performed using Two-tailed Student's _t_ test. **d** NTA analysis of EV levels released by double knockout mutants and wild type. Data are presented as mean values (±SD). Statistical analyses were performed using Two-tailed Student's _t_ test.

**e** Immunoblots of culture fluid from _P. sojae_ transformant over-expressing eGFP-PsTET3. The P100 (pellet) fraction and S100 (supernatant) fractions from the culture fluid were analyzed alongside a whole mycelium protein lysate. CBB staining was used as a loading control. **f** Trypsin digestion of _P. sojae_ EVs. The red star indicates the size expected for the N-terminal eGFP-containing fragment protected from trypsin. **g** EV sucrose gradient centrifugation and protein content analyzed by western blot. **h** Localization of eGFP-PsTET3 during infection of soybean hypocotyl epidermal cells by an eGFP-PsTET3 overexpressing _P. sojae_ transformants. White arrows indicate putative EVs. The control transformant expresses eGFP in the _P. sojae_. Scale bars, 10 μm. The brightness of all images was increased identically to enhance visualization of EVs. **i** Confocal microscopy and fluorescence analysis of EVs isolated from _P. sojae_ expressing eGFP-PsTET3 with or without FM4-64. Scale bars were indicated. All experiments were repeated three times with similar results. Source data are provided as a Source Data file.

recombinant purified PsTET3 EC2-His, we examined the phosphorylation of mitogen-activated protein kinases (MAPKs) in _N. benthamiana_ leaves by immunoblotting using anti-P42/44 antibodies. Recombinant purified EC2-His triggered strong MAPK phosphorylation compared to treatment with EC1-eGFP-His. However, MAPK phosphorylation was much weaker in _NbSERK3a/b_-silenced _N. benthamiana_ leaves than in control plants (Fig. 4f). EC2-His protein also triggered plant immunity in soybean, including a ROS burst, upregulated _PATHOGENESIS-RELATED 1_ (_PR1_) gene expression, and MAPK phosphorylation (Fig. 4g, h and Supplementary Fig. 14f). Together, these results demonstrate that EC2

is the key region of PsTET3 recognized by both _N. benthamiana_ and soybean.

## The 16 C-terminal amino acids of EC2 are required for the recognition of PsTET3 by _N. benthamiana_

To define the region of EC2 responsible for plant responses to PsTET3, we generated truncated versions of PsT3M4 (which retained activity). Beginning with the 20 amino acids deleted from PsT3M4, we generated five successive deletions from the N terminus of the remaining EC2 region: D1, D2, D3, D4, and D5 (Fig. 5a and Supplementary Data 3).

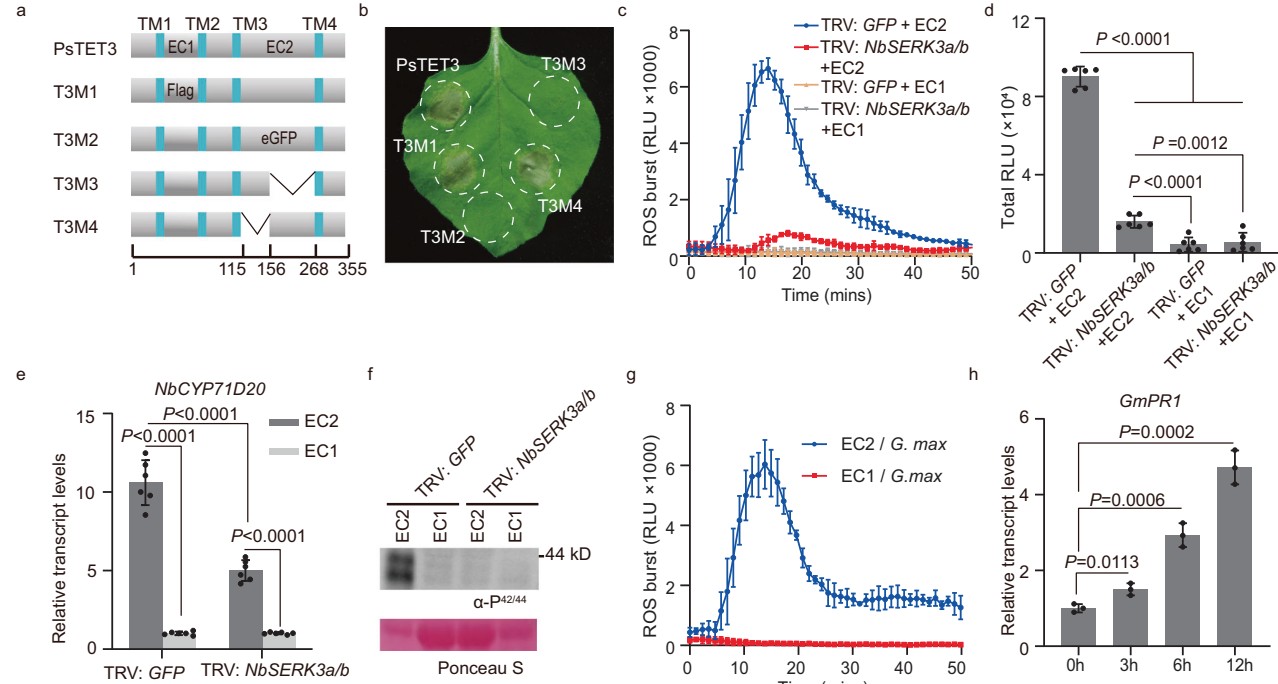

**Fig. 4 | EC2 is the key region for PsTET3 to induce immune responses.**
**a** Schematic diagram showing the protein structures of PsTET3 and derived dele-tion or replacement mutants. PsTET3 and indicated mutants fused with eGFP at the N terminus. **b** Representative *N. benthamiana* leaves infiltrated with indicated constructs were photographed 3 days after infiltration. **c**, **d** Production of ROS in TRV: *GFP* or TRV: *NbSERK3a/b N. benthamiana* leaf discs treated with 1 μM EC2-His protein or control. Mean values (±SD) of six measurements are shown. Statistical analyses were performed using Two-tailed Student's *t* test. **e** Relative transcript levels of PTI-marker gene *NbCYP71D20* in TRV: *GFP* or TRV: *NbSERK3a/b N. ben-thamiana* leaves infiltrated with 1 μM purified EC2-His protein or control (EC1-eGFP-His). Levels of *NbCYP71D20* were normalized to *NbEF1α* then set relative to the levels of the buffer control (which was set to 1). Mean values (±SD) of six

measurements are shown. Statistical analyses were performed using Two-tailed Student's *t* test. **f** MAPK phosphorylation triggered by 1 μM EC2-His in TRV: *GFP* or TRV: *NbSERK3a/b N. benthamiana* leaves. Ponceau S-stained Rubisco protein is shown as a total protein loading control. **g** Production of ROS in *G. max* leaf discs treated with 1 μM EC2-His protein or control. Mean values (±SD) of three replicates are shown. **h** Relative transcript levels of *PR1* gene in *G. max* roots treated with 1 μM purified EC2-His protein or control (EC1-eGFP-His). Relative transcript levels were determined by RT-qPCR. Mean values (±SD) of three replicates are shown. Statis-tical analyses were performed using Two-tailed Student's *t* test. All experiments were repeated three times with similar results. Source data are provided as a Source Data file.

Variant D5, comprising only 12 amino acids, no longer caused cell death, whereas D4, comprising 32 amino acids, still induced cell death (Fig. 5a). We deleted 4, 8, 12, or 16 amino acids from the N terminus or C terminus of the remaining EC2 region of D4, producing 11 additional deletion variants (Fig. 5a and Supplementary Data 3). When we expressed each construct encoding these variant proteins in *N. benthamiana*, the smallest region that still induced cell death was D9, comprising 16 amino acids at the C terminus of EC2 (Fig. 5a). Ion leakage assays produced similar results (Fig. 5a). All constructs were well expressed in *N. benthamiana* (Supplementary Fig. 15a). However, the synthetic 16–amino acid peptide did not show ROS-induced ability in *N. benthamiana* leaves (Supplementary Fig. 15c). D5 and D9 were correctly targeted to the plasma membrane and apoplast (Supple-mentary Fig. 12), and D9 localized to *N. benthamiana* EVs (Supple-mentary Fig. 16). The cell death triggered by D9 still required NbSERK3a/b (Supplementary Fig. 17a, b).

We mutated these 16 amino acids to alanine (A) individually (Fig. 5b). When the substitution mutants were expressed in *N. benthamiana*, cell death was significantly reduced when the first, third, fifth, thirteenth, and fifteenth positions (E, R, V, W, and N, respectively) were mutated (Fig. 5b). Ion leakage assays confirmed this result (Fig. 5b). Immunoblotting showed that all mutant proteins accumulated (Supplementary Fig. 14b). These results suggest that the 16 C-terminal amino acids of EC2 are required for PsTET3 to trigger plant immune responses and that five specific amino acids are required for this activity, either for direct recognition or for the structure of EC2.

## Sequence differences in the C terminus of EC2 enable *N. benthamiana* to distinguish self from non-self tetraspanins

Considering the widespread distribution of TET proteins in eukar-yotes, we wondered whether TET proteins from other kingdoms could trigger cell death in *N. benthamiana* leaves. To explore this issue, we mined the genomes of 8 oomycetes, 13 fungi, and 7 plant species for *TET* genes. We identified 95 genes encoding TET proteins and sub-jected them to phylogenetic analysis (Fig. 6a). In the phylogenetic tree, TET protein sequences from plants, fungi, and oomycetes were well separated from each other (Fig. 6a), which is consistent with inde-pendent radiation within each kingdom.

In oomycetes, we identified three to five *TET* genes in each species. We cloned all *TET* genes from oomycete species (including *P. sojae*, *Phytophthora infestans*, *Phytophthora capsici*, *Phytophthora parasitica*, *Pythium ultimum*, and *Pythium oligandrum*) and expres-sed them in *N. benthamiana*. For each oomycete pathogen, two to four TET proteins induced cell death in *N. benthamiana* (Fig. 6a and Supplementary Fig. 18a, b). All TET proteins in the clades containing PsTET1 and PsTET3 induced cell death in *N. benthamiana* (Fig. 6a). The sequence information of PsTETs were provided in Supplemen-tary Data 4.

In fungi, only the Pls1 family of *TET* proteins is present in both ascomycetes and basidiomycetes, including both pathogenic and non-pathogenic species[19]. We cloned the *Pls1* genes from fungal pathogens with different lifestyles, including the obligate biotroph *Phakopsora pachyrhizi* (a basidiomycete) and three ascomycetes: the wilt pathogens *Verticillium dahliae* and *Fusarium oxysporum* and the hemi-biotroph

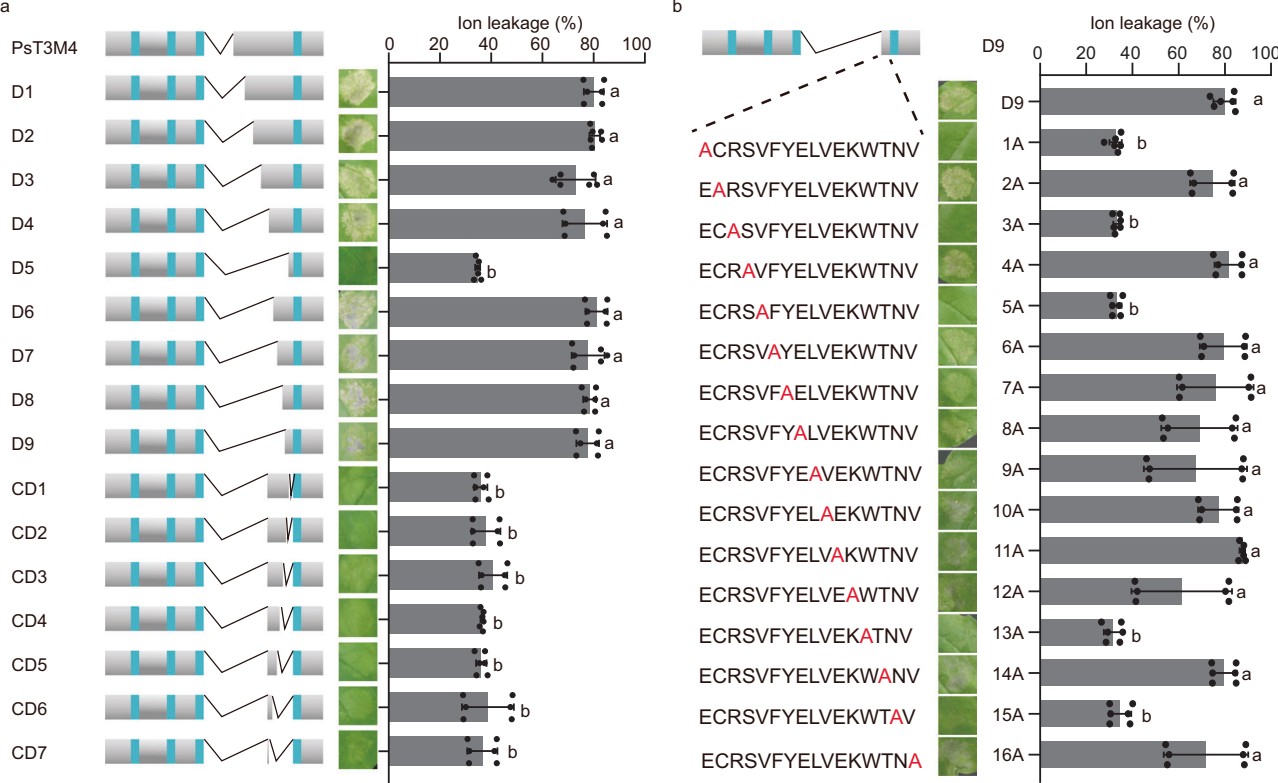

**Fig. 5 | The 16 residues at the C-terminus of EC2 are required to induce cell death in *N. benthamiana*. a** Schematic diagram showing the construction of PsT3M4 and derived deletion mutants. The letter D indicates deletions from the remaining N-terminus of EC2 and the letters CD indicate deletions from the EC2 C-terminus. Residues deleted in each mutant are listed in Supplementary Data 3. *N. benthamiana* leaf panels infiltrated with indicated constructs were photographed 3 days after infiltration. Representative leaves are shown. Quantification of cell death was performed using an ion leakage assay at 3 dpi. Mean values (±SD) of six measurements are shown. Different letters represent significant differences (P < 0.0001; one-way ANOVA). **b** Schematic diagram of the mutant D9 and the single amino acid substitution mutants of D9. *N. benthamiana* leaves infiltrated with indicated constructs were photographed 3 days after infiltration. Quantification of cell death using ion leakage after 3 dpi. Mean values (±SD) of six measurements are shown. Different letters represent significant differences (P < 0.0001; one-way ANOVA). For exact *p* values, see source data. All experiments were repeated three times with similar results. Source data are provided as a Source Data file.

*Magnaporthe oryzae*. All four cloned genes induced cell death in a NbSERK3a/b-dependent manner when expressed in *N. benthamiana* (Fig. 6a and Supplementary Fig. 19a, b). As observed in PsTET3, the 16 C-terminal residues of EC2 were also required for *P. pachyrhizi* Pls1 (PpaTET) to induce cell death in *N. benthamiana* (Supplementary Fig. 20a–c). PpaTET was correctly targeted to the plasma membrane, apoplast, and EVs of *N. benthamiana* (Supplementary Figs. 12, 16). We confirmed that the cell death triggered by the 16 C-terminal residues of PpaTET (PpaTET-16aa) still required NbSERK3a/b in *N. benthamiana* (Supplementary Fig. 17c, d).

Plants contain more than 10 *TET* genes. We cloned all 10 predicted TET genes from soybean, four from *N. benthamiana*, and two each from Arabidopsis, rice (*Oryza sativa*), cotton (*Gossypium hirsutum*), potato (*Solanum tuberosum*), and tomato (*Solanum lycopersicum*), with orthologs for *AtTET8* and *AtTET9* in each species. None of the plant *TET* genes induced cell death in *N. benthamiana* leaves (Fig. 6a and Supplementary Fig. 21a, b). We swapped the EC2 domain of NbTET6 with that of PsTET3. NbTET6 containing the PsTET3 EC2 (NbT6M1) induced cell death in *N. benthamiana*, whereas PsTET3 containing the NbTET6 EC2 (PsT3M5) did not (Fig. 6c, d), further confirming that EC2 is the key region of PsTET3 that induces immune responses. Both NbTET6 and NbT6M1 were correctly localized (Supplementary Figs. 12, 16). We confirmed that the cell death triggered by NbT6M1 still required NbSERK3a/b (Supplementary Fig. 17e, f). To validate this finding, we replaced the 16 C-terminal EC2 residues of GmTET8 with those from PsTET3 (GmTET8-16aa). GmTET8 and GmTET8-16aa accumulated to adequate levels, but only GmTET8-16aa

induced cell death in *N. benthamiana* (Fig. 6e, f). When the five key residues of EC2 were mutated (GmTET8-16aaM), the variant protein no longer induced cell death (Fig. 6e, f). In addition, the cell death triggered by GmTET8-16aa still required NbSERK3a/b (Supplementary Fig. 17e, f).

To search for amino acids that might account for the specific recognition of microbial TETs, we aligned the last 16 residues of the EC2s from 6 plant TETs, 5 oomycete TETs, and 4 fungal TETs (Fig. 6b). The plant TETs included the EV proteins TET8 and TET9 from Arabidopsis[10], together with their soybean and rice orthologs. The oomycete TETs included the three *P. sojae* TETs that trigger cell death together with their *P. ultimum* orthologs. The fungal TETs included all four Pls1 proteins experimentally shown here to trigger cell death. Of the five positions shown in Fig. 5b to be required for the PsTET3 variant D9 to trigger cell death, three corresponded to sites where the oomycete and fungal residues consistently differed greatly in size and/ or charge from the plant residues. The fungal residues differed greatly from the plant and oomycete sequences at a fourth site.

Based on the above results, we suggest that *N. benthamiana* can recognize specific non-self tetraspanins from oomycete and fungal pathogens, while avoiding recognition of self-tetraspanins, by detecting sequence differences in the 16 C-terminal amino acid residues of EC2.

### Tetraspanins are key factors of EVs released by *P. sojae* to activate plant innate immunity

To determine whether *P. sojae* EVs trigger plant immune responses, we assayed ROS bursts in *N. benthamiana* leaves following treatment

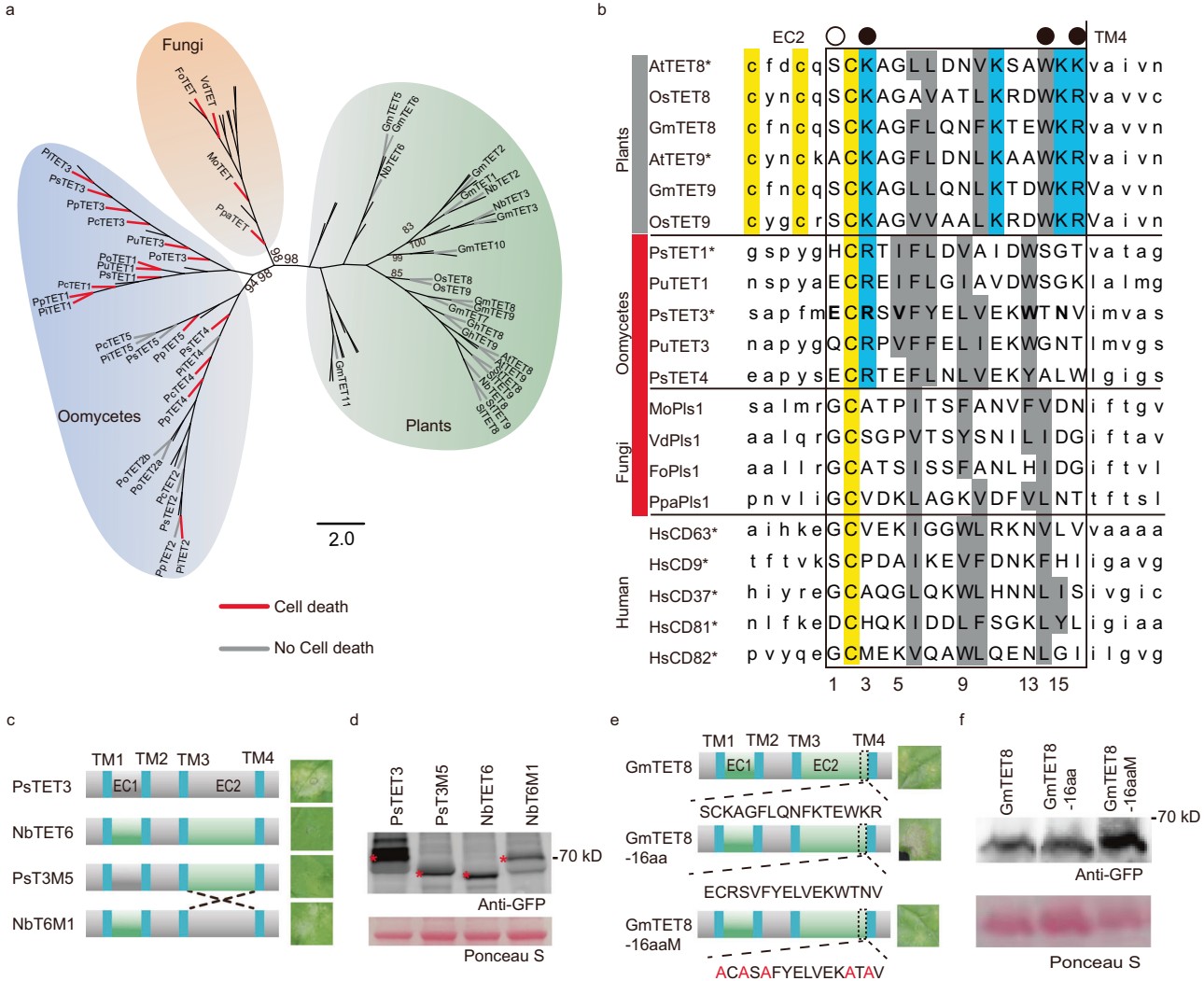

**Fig. 6 | TET proteins from oomycetes and fungi but not plants trigger cell death in *N. benthamiana*. a** A phylogenetic cladogram of 95 full-length tetraspanin family protein sequences originating from plants, fungi and oomycetes. The red or gray branches indicate the TET proteins that trigger cell death or not in *N. benthamiana*. **b** Alignment of amino acid sequences from the C-termini of plant, microbial and human EC2 regions. The final 16 residues of each EC2 are shown in upper case and enclosed by the box. PsTET3 residues required for the PsTET3 D9 mutant to trigger cell death are shown in bold. Black dots above residues 5, 13, and 15 indicate residues required for PsTET3 D9 cell death that differ greatly in size and/or charge between the microbe and plant sequences. White dot indicates residue 3 is required for PsTET3 D9 cell death and differs greatly in size and/or charge between the fungal and plant sequences. Cysteine residues are highlighted in yellow, conserved positively charged residues are highlighted in cyan, conserved large hydrophobic residues are highlighted in gray. Gray bars indicate TETs without cell death, while red bars indicate TETs with cell death, and unmarked TETs were not tested.

Asterisks indicate TETs experimentally shown to associate with EVs. **c** EC2 domain swap experiments. Exchange of EC2 between PsTET3 and NbTET6 produced mutants PsT3M5 and NbT6M1. Cell death triggered by the constructs was assessed 3 days after agro-infiltration. **d** Immunoblot analysis of transiently expressed PsTET3, NbTET6 and mutants. Ponceau S-stained Rubisco protein is shown as a total protein loading control. Red stars indicate expected sizes. **e** Replacement of the C-terminal 16 residues of EC2 of GmTET8 with those from PsTET3. GmTET8-16aa indicates GmTET8 carrying the 16aa from PsTET3. GmTET8-16aaM indicates a non-cell death mutant of GmTET8-16aa. All constructs carried an N-terminal eGFP tag. **f** Immunoblot analysis of transiently expressed eGFP-GmTET8, eGFP-GmTET8-16aa, and eGFP-GmTET8-16aaM in *N. benthamiana* leaves 2 days after agro-infiltration. Ponceau S-stained Rubisco protein is shown as a total protein loading control. All experiments were repeated three times with similar results. Source data are provided as a Source Data file.

with EVs from wild-type *P. sojae* or the double knockout mutants dKO-3 and dKO-37 using a luminol-based assay. The ROS bursts induced in *N. benthamiana* leaves by EVs from dKO-3 and dKO-37 were significantly lower, although they were still detected compared to wild-type EVs (Fig. 7a, b). Trypsin can degrade proteins outside of EVs without destroying the membrane structure (Supplementary Fig. 22a, b). Intriguingly, the ROS burst induced by EVs pretreated with trypsin was significantly decreased (Supplementary Fig. 22c, d). Also, the EV-induced ROS burst was partially dependent on NbSERK3a/b (Supplementary Fig. 22e, f). These results indicate that EV-associated TET proteins are important for EV-induced plant responses.

We further investigated the functions of EVs in *P. sojae*, finding that EVs induce ROS bursts in a dose-dependent manner (Supplementary Fig. 22g). To avoid inducing plant immunity, we infiltrated $10^6$ particles ml⁻¹ EVs into *N. benthamiana* leaves, followed by inoculation with *P. capsici*. Low concentrations of EVs promoted infection by *P. capsici* (Supplementary Fig. 22h, i).

However, TET proteins from plants cannot induce plant immunity according to the results in Fig. 6. Therefore, we isolated EVs from soybean and *N. benthamiana* (Supplementary Fig. 23a–c) and infiltrated them into *N. benthamiana* leaves to determine whether plant-derived EVs can induce plant immunity. We observed cell death only in *N. benthamiana* leaves infiltrated with *P. sojae* EVs (Fig. 7c). We also

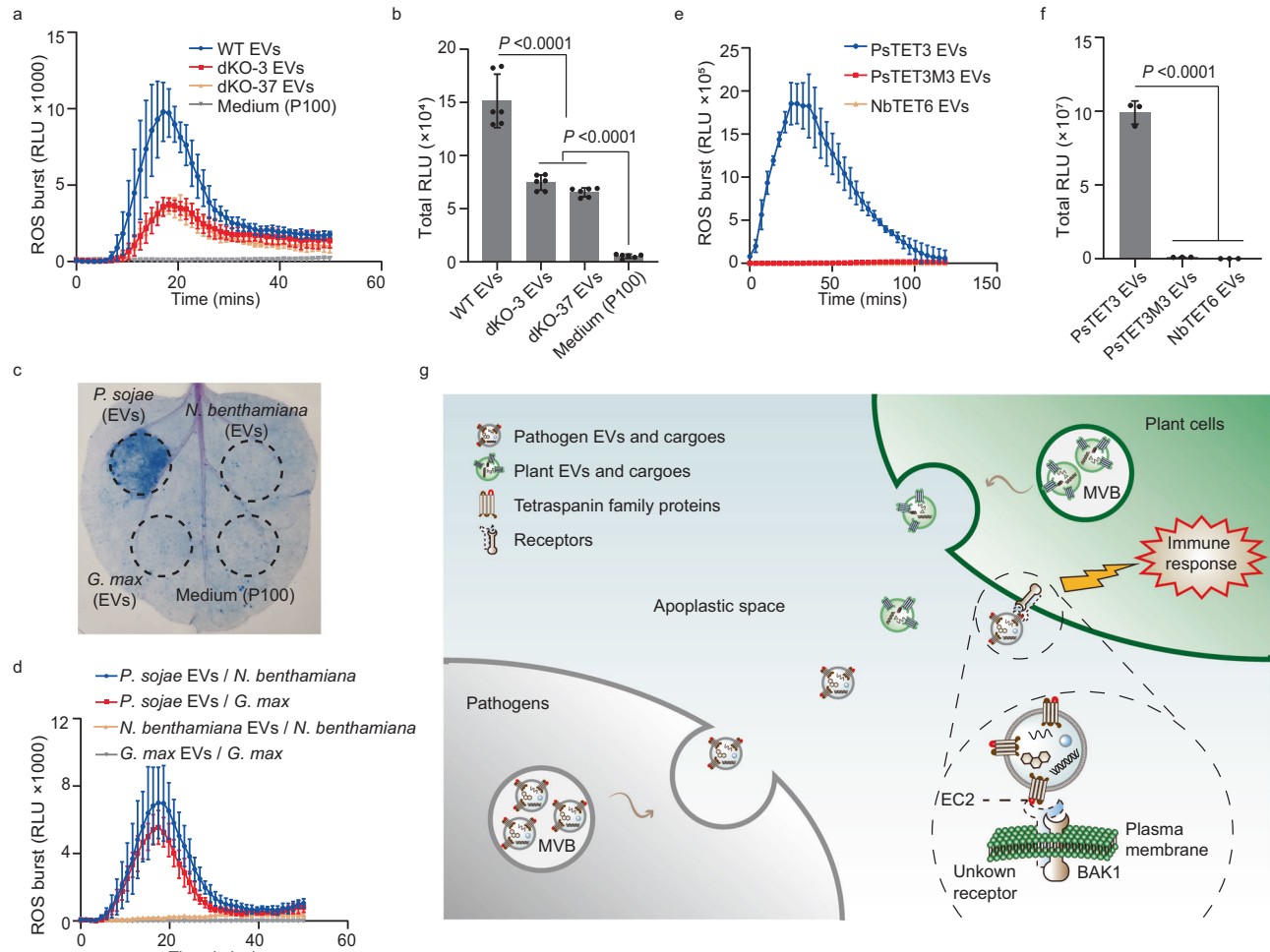

**Fig. 7 | EVs released by *P. sojae* can induce plant immune responses.**
**a**, **b** Production of reactive oxygen species (ROS) by *N. benthamiana* leaf discs treated with purified EVs ($10^8$ particles mL$^{-1}$) from *P. sojae* wild type and double knock out mutants. Mean values (±SD) of six measurements per experiment are shown. Statistical analyses were performed using Two-tailed Student's *t* test. RLU (Relative Luminescence Unit). **c** Representative leaves showing cell death triggered by *P. sojae* EVs in *N. benthamiana*. EVs ($10^8$ particles mL$^{-1}$) from *G. max* and *N. benthamiana* do not induce cell death. Medium P100 was used as negative controls. Leaves were stained with trypan blue and photographed after 3 dpi. **d** Production of ROS by *N. benthamiana* and *G. max* leaf discs treated with purified EVs ($10^8$ particles mL$^{-1}$) from *P. sojae*, *N. benthamiana* or *G. max*. Mean values (±s.e.m.) of three measurements per experiment are shown. RLU (Relative Luminescence Unit). **e**, **f** Production of ROS by *G. max* leaf discs exposed to purified EVs ($10^8$ particles mL$^{-1}$) from *N. benthamiana* after expression PsTET3, PsTET3M3 or NbTET6. Mean values (±SD) of three measurements per experiment are shown. Statistical analyses were performed using Two-tailed Student's *t* test. All experiments were repeated three times with similar results. Source data are provided as a Source Data file. **g** Model showing plant recognition of EVs released by microbes into the apoplastic space triggering innate immunity. MVB indicates multivesicular body.

detected ROS bursts in *N. benthamiana* and soybean leaf discs after treatment with purified EVs from *P. sojae* but not with EVs from plants (Fig. 7d). Finally, we isolated EVs from *N. benthamiana* transiently expressing *PsTET3*, *PsTET3M3* (which did not induce cell death), or *NbTET6*. Only EVs transiently expressing *PsTET3* induced ROS bursts in soybean leaves (Fig. 7e, f). These results indicate that *N. benthamiana* and soybean leaves can distinguish non-self from self EVs to trigger defense responses by specifically recognizing microbe-derived tetraspanins (Fig. 7g).

## Discussion

EVs are universal mediators of intercellular communication across numerous kingdoms of life[1]. In recent years, EVs have emerged as key players in cross-kingdom plant-microbe interactions. For example, pathogens secrete EVs that carry virulence factors to promote infection[12,13,33]. Despite increasing evidence for the importance of EVs in microbial infection, how plants respond to microbial EVs to mount defense responses remains unclear. Microbe-associated molecular patterns (MAMPs) or microbial effectors are perceived by pattern

recognition receptors within the complex extracellular milieu to activate the appropriate immune responses. In this study, we determined that microbe-derived EVs can also be recognized by plants including soybean and *N. benthamiana*. We identified two EV-localized proteins derived from *P. sojae*, PsTET1 and PsTET3, which triggered immune responses including cell death and a ROS burst in *N. benthamiana*. *N. benthamiana* responded to many TET proteins from oomycete pathogens, especially orthologs of PsTET1 and PsTET3, as well as fungal TET proteins in the Pls1 family (Fig. 6a). By contrast, *N. benthamiana* did not respond to any plant TET protein, including the EV-associated proteins AtTET8 and AtTET9 or their orthologs (Fig. 6a). The cell death triggered by key TET constructs used throughout the study, including fungal TETs, required the receptor-like kinase BAK1, a central regulator of plant PTI (Supplementary Figs. 17, 19), ruling out non-specific causes of cell death in each case.

Many MAMPs recognized by plant pattern recognition receptors are small peptide fragments of larger microbial proteins[28,34,35]. Here, we discovered that the last 16 amino acid residues of the EC2 of

PsTET3 are necessary to trigger plant immune responses (Fig. 5). The EC2 region alone (fused to a His tag) induced a ROS burst in both *N. benthamiana* and soybean tissues (Supplementary Fig. 14b and Fig. 4g), and the ROS burst in *N. benthamiana* required NbSERK3a/b (Fig. 4c). A EC2-His fusion protein also induced the transcription of three plant defense genes in *N. benthamiana* (Fig. 4e and Supplementary Fig. 14d, e). The ability of EC2 to induce immune responses in soybean as well as *N. benthamiana* suggests that a wider array of plants can detect oomycete TETs.

Alignment of the 16 C-terminal residues of EC2 from proteins experimentally demonstrated to be associated with EVs (PsTET1, PsTET3, AtTET8, AtTET9, HsCD9, HsCD37, HsCD63, HsCD81, and HsCD82) (Fig. 6b) revealed that all share a conserved cysteine residue at position 2 within this region, as well as alternating patches of hydrophobic and hydrophilic residues characteristic of an amphipathic alpha helix[29,30]. The four fungal Pls1 proteins also share these features, which is consistent with our observation that PpaTET was targeted to EVs in *N. benthamiana* (Supplementary Figs. 12, 16).

Domain swap experiments (Fig. 6c–f) confirmed that sequence differences in the 16-residue C-terminal peptide of EC2 were responsible for the lack of recognition of plant TETs. The results from mutagenesis of this peptide (Fig. 5) and sequence comparisons among plant, oomycete, and fungal sequences (Fig. 6b) suggested that positions 5, 13, and 15 might be involved in the recognition of non-plant TETs, as these three positions were required for the recognition of PsTET3 and differed greatly in size and/or charge between oomycetes/fungi and plants. Our current data do not reveal whether these positions are involved in TET-receptor contact, whether they are responsible for changes in EC2 structure that alter recognition, or both. Mutagenesis also indicated that positions 1 and 3 were required for the recognition of the PsTET3 D9 deletion mutant. However, position 1 is not well conserved, even among oomycetes, and position 3 in plants (conserved lysine) is highly similar to that in oomycetes (conserved arginine), but not in fungi. It remains to be determined if positions 1 and 3 are required for the recognition of full-length PsTET3. We also cannot rule out the possibility that different receptors with different specificities are required for the recognition of oomycete and fungal TET proteins. It would also be interesting to determine if plants can detect TETs from plant-associated microfauna such as nematodes and aphids.

EVs have emerged as key weapons for both hosts and pathogens. Hosts use EVs to target a wide array of defense compounds towards pathogens[1]. Plants use EVs to transport small RNAs into fungi to suppress infection[10,36]. Pathogens also promote infection by secreting EVs carrying virulence factors[12,13,33]. Of particular interest is the potential ability of EVs to deliver microbial RNAs and proteins such as effectors inside host cells to modify their normal physiological and biochemical functions to favor the microbe. *P. sojae* secreted eGFP-PsTET3-labeled EVs during infection progression, and some of these EVs appeared to be taken up by plant cells. In addition, we detected a few *P. sojae* RxLR effector proteins in the EVs. Plant-derived EVs labeled with eGFP were previously shown to be taken up by pathogen cells[9,10]. Microbial EVs enter animal cells via five different pathways: macropinocytosis, clathrin-mediated endocytosis, caveolin-mediated endocytosis, lipid raft-mediated endocytosis, and direct membrane fusion[37]. It remains to be determined which mechanisms are used by oomycete and fungal EVs to enter plant cells.

TET proteins, which are widespread in eukaryotes[38], interact with many membrane proteins, cytoplasmic proteins, and lipids, forming a functional unit called a tetraspanin-enriched microdomain[18,39]. Many proteins are sorted into EVs by interacting with TET proteins. For example in humans, the metalloproteinase CD10 is sorted into EVs by interacting with CD9[40]. In the current study, we detected the eGFP-PsTET3 fusion protein in EVs enriched by centrifugation at 100,000 $g$. Thus, TET family proteins such as PsTET3 may be used as protein

markers of EVs in oomycetes, like in animals and plants. Furthermore, our results demonstrate that oomycete and fungal TETs can be targeted to plant EVs, potentially providing a class of artificial biomarkers for EV research. EV secretion was significantly reduced in Arabidopsis *TET8* knockout mutants[41]. Likewise, a *P. sojae* double knockout mutant for the functionally redundant genes *PsTET1* and *PsTET3* showed substantially lower EV production and virulence. Knockout of Pls1 family genes in fungi resulted in the failure to form penetration pegs, leading to a significant decrease in pathogenicity[31,42–44]. Our results suggest that one function of fungal Pls1 family tetraspanins is to serve as components of fungal EVs.

Many marker proteins that can distinguish different subpopulations of EVs in animals have been identified. However, in plants, oomycetes, and fungi, fewer markers are currently available. PENETRATION1 (PEN1)-positive EVs and TET8-positive EVs have been identified in plants. Although we successfully isolated *P. sojae* EVs, our data do not reveal whether distinct sub-types of EVs carry PsTET1 or PsTET3 or whether there is additional structural or functional heterogeneity among *P. sojae* EVs. It is important to identify additional marker proteins, isolate and purify subsets of EVs, analyze the cargoes carried in these EVs, and study their functions in future studies.

## Methods

### Plants and *Phytophthora* cultivation

*Nicotiana benthamiana* and *Glycine max* plants were cultivated in growth chambers. Temperatures were set at 22-25 °C, and the light/dark period was 16 h/8 h. *P. sojae* wild-type strain P6497 and transformants and *P. capsici* wild-type strain LT1534 were grown in 10% V8 juice medium at 25 °C. For extraction of EVs from growth medium, *P. sojae* mycelia were cultured in liquid synthetic medium (1 L medium contained 0.5 g KH$_2$PO$_4$, 0.5 g yeast extract, 0.25 g MgSO$_4$·7H$_2$O, 0.001 g thiamine, 25 g glucose, 1 g asparagine, 0.01 g β-sitosterol). DNA transformations of *P. sojae*, and virulence assays were performed as described previously[45].

### CRISPR-Cas9-mediated gene knockouts in *P. sojae*

Gene deletion mutants were generated using the CRISPR-mediated gene replacement strategy[27]. The *mCherry* gene, ligated with two 1.0-kb fragments flanking the target gene, was used as donor DNA for homology-directed recombination. For double knockout mutants, we began with the single knock out mutants. After culturing the single knock out mutants for at least three generations in antibiotic-free plates for more than 2 weeks, the mutants could no longer grow on G418 antibiotic plates[45,46]. Using a single knock out mutant of *PsTET1* that had lost the resistance to G418 (KOT1-9), we conducted another CRISPR-mediated gene replacement to knock out the other gene *PsTET3* as described[45,46].

### Plasmid construction

Transmembrane protein-coding regions, including *PsTET3* and other tetraspanin homologs were cloned from the cDNA of the corresponding pathogens or plants. Each purified cDNA fragment was ligated to pBINGFP2 using the ClonExpress II One Step Cloning Kit (Vazyme Biotech Co. Ltd., Nanjing, China) to create an N-terminal eGFP fusion for expression in *N. benthamiana*. The *PsTET3* coding region was ligated into pTOReGFP to create an N-terminal eGFP fusion for expression in *P. sojae*. The fragment of *PsTET3* encoding the entire EC2 from position 116 to 268 was ligated into pPIC9K to create a C-terminal His-tag fusion for expression in *Pichia pastoris*. We employed the Tobacco rattle virus (TRV) RNA2 vector for gene silencing. This vector allows for the insertion of gene-silencing fragments[47]. We utilized this TRV2 vector to silence the NbBAK1/NbSERK3a/b genes in our experiments. Individual colonies for each construct were tested for inserts by PCR and selected clones were verified by sequencing.

## Transient expression in *N. benthamiana*

The constructs were transferred into *Agrobacterium tumefaciens* GV3101 by electroporation, and verified by PCR. LB liquid medium (with Kanamycin 50 mg L$^{-1}$, Rifampin 50 mg L$^{-1}$) was used to culture each strain at 28 °C with shaking at 200 rpm. After 18 h, the bacteria were collected by centrifugation at 4000 $g$ and resuspended three times in infiltration buffer (10 mM MgCl$_2$, 10 mM MES pH 5.7, 20 nM acetylsyringone). Finally, the suspension was adjusted to an OD$_{600}$ of 0.4. Cells carrying a P19 silencing suppressor gene were added to a 1:1 ratio, followed by infiltration into *N. benthamiana*. Protein expression was assayed by western blot analysis 2 days after infiltration. Cell death symptoms were observed and photographed 3–5 days after infiltration.

## Ion leakage assay

To quantify the degree of cell death in *N. benthamiana* leaves, ion leakage assays were performed. Three days after *Agrobacterium* infiltration, 6 leaf discs (diameter 8 mm) were taken from each sample and floated on 5 ml of distilled water for 3 h at room temperature. Then, the conductivity of the bathing solution was measured with a conductivity meter (S470 SevenExcellence; Mettler Toledo, Shanghai, China) to yield 'value A'. The leaf discs were then returned to the bathing solution and boiled in sealed tubes for 25 min. After cooling the solution to room temperature, the conductivity was measured again to obtain 'value B'. Ion leakage was expressed as the percentage of total ions, that is (value A/value B) × 100. All assays were repeated three times.

## Vesicle isolation and NTA analysis

For EV isolation from *P. sojae* culture fluid, mycelia were grown in 2 L synthetic liquid medium for 10 days. To obtain the culture filtrate, firstly the mycelium was removed by filtration with Miracloth (EMD Millipore Corp). Then, mycelial fragments and larger vesicles were further removed by centrifugation for 30 min at 10,000 $g$. Next, impurities were removed by filtration of the supernatant through a 0.22 μm membrane. Then the culture filtrate was concentrated to 100 ml by ultrafiltration through a cup with a molecular weight cutoff of 100 kd. Finally, EVs were sedimented by centrifugation at 100,000 g (P100), and resuspended in PBS, ready for western blot or mass spectrometry analysis. EVs were stored short-term at 4 °C and long-term at −80 °C. Diameters and concentrations of the particles were measured by Nanoparticle Tracking Analysis using ZetaVIEW S/N 252 (Software ZetaView 8.04.02 SP2). EVs of *N. benthamiana* and *G. max* are isolate from apoplastic wash fluids of leaves. The isolation of EVs from *N. benthamiana* and *G. max* was performed as described previously[48], and 100,000 $g$ was used to isolate the EVs.

## Sucrose gradient separation of EVs

The sucrose gradient separation of EVs was as described previously[36]. 10–85% sucrose stocks (w/v) were prepared, including 10%, 15%, 20%, 25%, 30%, 35%, 40%, 45%, 50%, 55%, 60%, 70%, 75%, 80% and 85%. For top loading, 500 μl 10% sucrose was used to resuspended P100 fraction. The total volume is 16.5 ml. Samples were centrifuged in a swinging-bucket rotor for 16 h at 100,000 $g$, 4 °C and six fractions (2.75 ml each) were collected.

## Mass spectrometry

EVs were isolated from culture filtrate of *P. sojae* P6497 strain. After adding SDT buffer to the sample, the lysate was sonicated and then boiled for 15 min. And then, following centrifugation at 14,000 $g$ for 40 min, the supernatant was quantified using the BCA Protein Assay Kit (Bio-Rad, USA)[49].

After washing with UA buffer (8 M Urea, 150 mM Tris-HCl pH 8.0), 100 μl of iodoacetamide (100 mM IAA in UA buffer) was added to block reduced cysteine residues, and the samples were incubated for 30 min in darkness. The filters were washed three times with 100 μl UA buffer

and twice with 100 μl 25 mM NH$_4$HCO$_3$. Afterward, the protein suspensions were digested overnight with 4 μg trypsin (Promega) in 40 μl 25 mM NH$_4$HCO$_3$ buffer at 37 °C, and the resultant peptides were collected. The peptides of each sample were desalted on C18 Cartridges (Empore™ SPE Cartridges C18 (standard density), bed I.D. 7 mm, volume 3 ml, Sigma), concentrated by vacuum centrifugation and reconstituted in 40 μl of 0.1% (v/v) formic acid. The peptide content was estimated by UV light spectral density at 280 nm using an extinction coefficient of 1.1 of 0.1% (g/l) solution that was calculated on the basis of the frequency of tryptophan and tyrosine in vertebrate proteins[49].

The peptide mixture was loaded onto a reverse phase trap column (Thermo Scientific Acclaim PepMap100, 100 μm*2 cm, nanoViper C18) connected to the C18-reversed phase analytical column (Thermo Scientific Easy Column, 10 cm long, 75 μm inner diameter, 3μm resin) in buffer A (0.1% Formic acid) and separated with a linear gradient of buffer B (84% acetonitrile and 0.1% Formic acid) at a flow rate of 300 nl/min controlled by IntelliFlow technology. For 2 h gradient, 0–55% buffer B for 110 min, 55–100% buffer B for 5 min, hold in 100% buffer B for 5 min. MS data were acquired using a data-dependent top10 method dynamically choosing the most abundant precursor ions from the survey scan (300–1800 $m/z$) for HCD fragmentation. The automatic gain control (AGC) target was set to 3e6, and maximum inject time to 10 ms. Dynamic exclusion duration was 40.0 s. Survey scans were acquired at a resolution of 70,000 at $m/z$ 200 and resolution for HCD spectra was set to 17,500 at $m/z$ 200, and the isolation width was 2 $m/z$. Normalized collision energy was 30 eV and the underfill ratio, which specifies the minimum percentage of the target value likely to be reached at maximum fill time, was defined as 0.1%. The instrument was run with peptide recognition mode enabled.

The MS data were analyzed using MaxQuant software version 1.5.3.17 (Max Planck Institute of Biochemistry in Martinsried, Germany)[50]. An initial search was set at a precursor mass window of 6 ppm. The search followed an enzymatic cleavage rule of Trypsin/P and allowed maximal two missed cleavage sites and a mass tolerance of 20 ppm for fragment ions. Enzyme = Trypsin, Missed cleavage = 2, Fixed modification: Carbamidomethyl (C), Variable modification: Oxidation(M), Decoy database pattern= Reverse. The cutoff of global false discovery rate (FDR) for peptide and protein identification was set to 0.01.

The MS data were searched against the *P. sojae* proteomic database (https://genome.jgi.doe.gov/portal/pages/dynamicOrganismDownload.jsf?organism=Physo1_1). All proteomes were categorized based on GO annotation. GO term data were from JGI (https://jgi.doe.gov/), and the analysis was carried out by TBtools v1.106 (GO enrichment module) following with Benjamini−Hochberg method for p-value adjustment. The proteins in *P. sojae* EVs identified in this study and the accession number are provided in Supplementary Data 1.

The mass spectrometry proteomics raw data have been deposited to the ProteomeXchange Consortium via the PRIDE[51] partner repository with the dataset identifier PXD040458 (ProteomeXchange Dataset PXD040458).

## Expression and purification of recombinant EC2 and EC1 protein

The *Pichia pastoris* strain KM71 was used for protein expression. Protein expression and purification were as described previously[52]. Protein was dissolved in buffer (20 mM Tris-HCL + 150 mM NaCl). There is a His tag fused in C-terminal of EC2 and an eGFP-His tag in C-terminal of EC1. For leaf infiltration experiments, the protein concentrations were adjusted to 1 μM in ddH$_2$O.

## Measurement of ROS production

5 mm diameter leaf discs were excised from 5-week-old *N. benthamiana* leaves or 2-week-old soybean leaves then placed in 96 well plates

with 200 μL sterile H$_2$O overnight. Next, the water was removed and 200 μl reaction solution (luminol 35.4 μg mL$^{-1}$, peroxidase 10 μg mL$^{-1}$ with 1 μM EC2 or 1 μM EC1) was added. When testing the ROS burst induced by EVs, an EV concentration of around 10$^8$ particles mL$^{-1}$ was used. The reaction solution is 200 ml (luminol 35.4 μg mL$^{-1}$, peroxidase 10 μg mL$^{-1}$ with 10$^8$ particles mL$^{-1}$ EVs or medium P100 fraction). Before treating the plants, the particle concentration must be adjusted to the same level by buffer according to NTA data. Luminescence was measured using a GLOMAX96 microplate luminometer (Promega, Madison, WI, USA).

## Quantitative PCR analysis of RNA and DNA

Total RNA was isolated using an RNA kit (Omega Bio-Ten, Norcross, GA, USA) and then used as template for cDNA synthesis. Reverse transcription was performed using HiScript II 1st Strand cDNA Synthesis Kit (Vazyme Biotech Co. Ltd., Nanjing, China). Genomic DNA was extracted using the genomic DNA kit (TIANGEN Biotech, Beijing, China) following procedures described by the manufacturer. qPCR was performed on an ABI 7500 Fast Real-Time PCR system (Applied Biosystems Inc., Foster City, CA, USA) using ChamQ SYBR Color qPCR Master Mix (Vazyme Biotech Co., Ltd, China). The primers used in this assay are listed in Supplementary Data 5. Data were analyzed using the 2$^{-\Delta\Delta CT}$ method.

## Immunoblots

Mycelial tissue or *N. benthamiana* leaves were fully ground into powder in liquid nitrogen. The powder was added to 1 ml lysis buffer [10 mM Tris-HCl (pH 7.5), 150 mM NaCl, 0.5% NP40, 0.5 mM ethylenediamine-tetraacetic acid (EDTA), plus a protease inhibitor cocktail (Sigma-Aldrich, St Louis, MO, USA)], vortexed fully, then placed on ice for 10 min. Next, centrifugation was performed at 4 °C for 10 min at 14000 g. For each lane, 40 μL supernatant was added to 10 μL SDS loading buffer, then boiled for 10 min before loading. For EV analysis, 40 μL of EV suspension (see above) was used. A standard SDS poly-acrylamide gel electrophoresis (SDS-PAGE) protocol was followed for protein separation. Then the proteins were transferred to Poly-vinylidene fluoride (PVDF) membranes. The membrane was blocked using 5% non-fat milk in PBST buffer (1 × PBS + 0.1% Tween 20) for 30 min at room temperature with 60 r.p.m. shaking. The antibodies (anti-GFP; 1:5000; Abmart #M20004) were then added to PBST with 2 h incubation at room temperature. Then the PVDF membranes were washed three times (5 min each) with PBST. The membranes were then incubated with goat anti-mouse at a dilution of 1:10000 in PBST at room temperature for 30 min with 60 r.p.m. shaking; and followed by three washes (5 min each) with PBST. The PVDF membranes were visualized using a scanner (Li-Cor Odyssey; Li-Cor Biotechnology, Lincoln, NE, USA) with excitation at 700 nm and 800 nm. When detecting MAPKs, TBST (1 × Tris-HCL + 0.1% Tween 20) was used instead of PBST. Uncropped scans of blots and gels are provided in Source Data file.

## Transmission electron microscopy

For transmission electron microscopy (TEM), 6 μL of vesicles suspended in 1 × PBS were applied to Formvar-carbon-coated copper grids. After blotting with filter paper, a drop of 2% phosphotungstic acid was placed on each copper grid. After 1–2 min, the dye solution was removed with filter paper. The grids were then rinsed with three successive drops of water. Finally, the grids were allowed to air dry, and then imaged at 100 kV using a HT7700 TEM.

## Confocal microscopy

For localization of PsTET3, *P. sojae* transformants expressing eGFP-PsTET3 were cultured in 10% V8 liquid medium. After 3 days, a small amount of mycelium was placed under a confocal microscope (LSM 710 laser scanning microscope, Carl Zeiss, Jena, Germany) to observe the fluorescence using 63× oil objective lens. To observe the uptake of

EVs by plant cells during infection, the hypocotyls of etiolated soybean seedlings were infected by zoospores of *P. sojae* transformants expressing eGFP-PsTET3. After 12 h, the epidermal cells within the infected area were observed using a ×20 objective lens. For the localization of TET proteins and the mutants in *N. benthamiana*, after 24 h following *A. tumefaciens* inoculation, the leaves discs were observed using a 20× objective lens. In each case, GFP fluorescence was observed at an excitation wavelength of 488 nm and an emission wavelength of 488–510 nm. For observing the eGFP-PsTET3 labeled EVs in vitro, the confocal microscope (ZEISS LSM 980 with Airyscan2) was used. GFP fluorescence was observed at an excitation wavelength of 488 nm and an emission wavelength of 509 nm. RFP fluorescence was observed at an excitation wavelength of 543 nm and an emission wavelength of 583 nm.

## Proteinase treatment

For the assay of ROS production by proteinase-treated EVs, trypsin was added to 40 μL EVs to a final concentration of 10 μg ml$^{-1}$. After incubation at 37 °C for 30 min, the EVs were recovered by 1.5 h centrifugation at 100,000 g. For protease protection assays, EVs (40 μL per treatment) were mixed with 10 μg ml$^{-1}$ trypsin with or without 1% Triton X-100. Then the samples were incubated in 37 °C for 30 min.

## Bioinformatic analysis

To search for tetraspanin family proteins in each genome, an HMM search was performed using the software HMMER (version 3.0; with default parameter) and a tetraspanin HMM (PFAM ID: PF00335; downloaded from http://pfam.xfam.org) against geno-mic databases downloaded from NCBI. The potential candidates were further validated by analysis with SMART (http://smart.embl-heidelberg.de), Pfam (http://pfam.xfam.org/), and TMHMM Server v. 2.0 (http://www.cbs.dtu.dk/services/TMHMM-2.0/). Previously published tetraspanin sequences were downloaded from NCBI. For the phylogenetic analysis, full-length tetraspanin protein sequences were aligned using the ClustalW2 program and a phylogenetic tree was constructed using MEGA7 with maximum likelihood. For the alignment of the EC2 C-terminal sequences, manual non-gapped alignment was used due to the shortness of the sequences.

## Statistics and reproducibility

Statistical analyses were performed using the software GraphPad Prism and Excel. All data are represented as the mean ± SD. The sample sizes were reasonable numbers for the statistical analysis used in this paper. No data have been excluded. The experiment design and experiment data collection are randomized. The infection assays were recorded in a blind way. The findings of all key experiments were reliably reproduced. Source data are provided as a Source Data file.

## Reporting summary

Further information on research design is available in the Nature Portfolio Reporting Summary linked to this article.

# Data availability

The data that support the findings of this study are available in the Supplementary Information. The mass spectrometry proteomics raw data have been deposited to the ProteomeXchange Consortium via the PRIDE partner repository with the dataset identifier PXD040458. Source data are provided with this paper.

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

## Acknowledgements

The authors thank Wenbo Ma (The Sainsbury Laboratory) for helpful suggestions. This research was supported by China Natural Science Foundation Grant 32020103012, China Funds for Innovative Research Groups Grant 31721004, the key program of National Natural Science Foundation of China Grant 31430073, and Chinese Modern Agricultural Industry Technology System Grant CARS-004-PS14.

## Author contributions

J.Z., Z.M., Yan Wang, and Yuanchao Wang designed the research. J.Z., Q.Q., Y.X., Y.S., F.L., and H.W. performed research. J.Z., Z.Z., H.S., W.Y., and S.D. analyzed the data. J.Z., Yan Wang, Z.M., and Yuanchao Wang wrote the paper.

## Competing interests

The authors declare no competing interests.
