## [Peer Review File · Nature Communications]

Divergent sequences of tetraspanins enable plants to specifically recognize microbe-derived extracellular vesiclesREVIEWER COMMENTS

Reviewer #1 (Remarks to the Author):

The manuscript NCOMMS-22-34996 by Zhu et al. is an interesting study that seeks to provide a better understanding of the biological roles of extracellular vesicles (EVs) produced by the oomycete pathogen *Phytophthora sojae* during plant infection. This is very challenging work, especially when trying to examine the cargo of the EVs and interaction with the plant's immune system. At this point I find that work, while potentially very exciting and contributing to our understanding of plant pathogen-derived EVs, the data appear too preliminary to be published without additional experiments. Primary concerns expressed were:

1. The authors don't use MISEV guidelines although performed the experiments using NTA, TEM and proteomic analysis. Please see <https://www.tandfonline.com/doi/full/10.1080/20013078.2018.1535750>
For example, PsEV isolation/characterization was performed from conditioned media. Precautions such as the presence of particles in the media and percent dead cells should be considered and indicated. This also raises the question whether "buffer" as in several experiments used, was an appropriate control. Furthermore, immunoblot analysis confirming the presence of EV marker proteins should also check for EV depleted proteins. Please also provide experimental details according to MISEV.
2. The authors don't use sucrose gradients or SEC to improve the purity of the isolated EVs, since P100 includes also larger protein aggregates. Along the same lines, as the TEM images show only individual vesicles, were other structures present in the images that could be contaminants or broken cells?
3. The authors don't use lipid staining of the EVs, e.g. in combination with NTA or to trace GFP-positive EVs in confocal microscopy. It is also unclear whether the authors use NTA to treat plants with similar EV concentrations. To this end, it would be interesting to observe the results using distinct EV concentrations.
4. Please indicate how many times the experiments were repeated independently with similar findings.

Figure 1: How relevant is the identification of 2 RxLR effectors in PsEVs? How does this compare to the overall secretome or proteome of Ps in this culture conditions? Did the authors identify any proteins in PsEVs that would provide insights into their biogenesis/release? Furthermore, given the focus on cell death inducing activities, it is unclear to me why data presented in Figure 7c and 7d were not included here.

Figure 2: Since the cell death inducing activity appears to depend on BAK1 (correct is NbSERK3a/b), it is unclear to me why PsTET1 and PsTET3 constructs without SP motif were not tested. In 2d it is unclear whether the P100 and S100 were collected from apoplastic wash fluids. Here, MISEV guidelines should also be considered.

Figure 3: It is surprising that the *P. sojae* PsTET1 and PsTET3 double mutants appear to have no general growth defects. Can the authors please comment. 3d, please include e.g. FM4-64 tracing to provide additional evidence of GFP-positive PsEVs at or in plant cells. Also, use isolated GFP-PsEVs and apply to plant cells. How relevant is a Ps control strain expressing cytosolic GFP?

Figure 7: Were similar PsEV concentrations applied when collecting EVs from wild type vs mutants? Do the strains produce EVs at similar concentrations? Would proteinase-treated and washed PsEVs still induce cell death?

Figure S4: Can the authors comment what the smaller and larger spots were? Also, GFP-PsTET3 appears to show a signal around the nucleus? And GFP-AtARA6 is plasma membrane while this should be late endosome?

Figure S5: Please provide data showing the knock-out of PsTET1 and PsTET3 using RT-PCR.

Figure S6: Since GFP-PsTET3 is overexpressed, does it result in more vesicle production and affect virulence?

Figure S9: Is buffer the best control given expression in yeast and yeast extract stimulating plant immunity?

Discussion: It remains unclear why RxLR effectors would be present in EVs, potentially delivered by EVs into plant cells, yet TET proteins present at EVs induce cell death?

Reviewer #2 (Remarks to the Author):

The authors provide a preliminary report on an EV proteome of *Phytophthora sojae*. They identify candidate tetraspanins (TETs) and report that these trigger cell death when expressed in *Nicotiana benthamiana* cells, assuming that they are also exported in EVs by the plant host to trigger PTI-associated cell death mediated by BAK1. They characterise the region of TET1 and TET3 that act as PAMPs to activate PTI and show that orthologous regions in plant TET equivalents are divergent and therefore not detected in a similar way. The work is, at best, preliminary and has a number of significant deficiencies that preclude publication.

Major Criticisms:

1. The description of EV purification is undetailed. It ultimately consists of a 100,000 xg ultracentrifugation spin to give a pellet (P100). There are no low-magnification images in Sup Fig 1. It is widely recognised that a pellet from a 100k spin will gather a great deal of membrane-associated material. As a result it is a requirement to resuspend and then perform a sucrose gradient to purify EVs.
2. The information on the proteome of the EVs is not acceptable! There are no raw data provided and the protein IDs are not publicly available in accepted international databases. It is thus not possible to cross-check what the proteins are. There is no proteome provided from the supernatant (i.e. conventionally secreted proteins) for comparison. I would expect such comparisons to include volcano plots, for example. There is no comparison of the 'EV' proteome to other EV studies.
3. The authors express candidate EV-associated transmembrane proteins in *N. benthamiana* in Fig 1, assuming they will be exported in plant EVs. They do not provide evidence using electron microscopy or confocal microscopy that these proteins are incorporated in *N. benthamiana* EVs. The evidence that TET proteins are in EVs is provided by a 100k spin (P100) from lysate, rather than apoplast purification (Fig 2d). The proteins could be in any membrane from inside the cell. The evidence in Suppl Fig S4 that the TET proteins are exported is extremely poor. There is no demonstration that these GFP-labelled proteins are exported using additional markers (e.g. membrane). GFP is degraded in the *N. benthamiana* apoplast. The AtAra6 used as a negative control in Fig 2d is a poor one. Ara6 will also be associated with early endosomes, which would be co-purified in the P100.
4. The growth rate is not provided for the double KO lines in Suppl Fig 5 – I cannot see growth over time.
5. In Fig 3a 5 TET family members are described. A brief search of the *P. sojae* genome indicates there are 6 family members.
6. For Fig 3f I would like to see that triton has disrupted the integrity of the EVs, using TEM and nanoparticle tracking, at least. Also, show that trypsin alone does not disrupt the integrity of the EVs.
7. In Fig 3g, provide convincing evidence that the green dots are indeed EVs. This is a very preliminary result.
8. Expression of TET mutants (Suppl Fig 8) and targeting to the *N. benthamiana* membrane and export into the apoplast (Suppl Fig 4) are not adequately demonstrated in my view. The immunoblots are very poor, as are the confocal images (which have no markers in the plant to be associated with).
9. In Fig 6, why did they not synthesise the 16aa peptide from oomycetes to show that it triggers PTI cell death, whereas the plant equivalent does not?
10. The BAK1-dependent recognition of TET proteins is based upon expression of TET proteins, or their expression and infiltration, into plant leaves (Fig 4). *P. sojae* EVs also trigger cell death (Fig 7), but do they do so in a BAK1-dependent way?

Minor comments

Line 172 – reference the PTI marker genes.

Reviewer #3 (Remarks to the Author):

Jinyi Zhu and colleagues report on extracellular vesicles (EVs), released by the disastrous soybean pathogen *Phytophthora sojae* (Ps), and the plant recognition of EV-associated Ps tetraspanin (TET) PsTET3 that induces a plant cell death-like immune response in a BAK1-dependent manner. The role EVs in plant-pathogen interactions is a relative young research field and many novel discoveries are expected in future. Plant TETs have been previously reported as EV biomarkers. Here, a pathogen EV-TET is described for the first time as an immune trigger, therefore extending our knowledge on EV-TETs in plant-pathogen interaction. The manuscript is written well and major findings are displayed in a suitable manner.

Please, find my critical comments to improve the work as follows:

Figure S1: please, include a NTA measurement of only liquid medium control. This is crucial, because the medium contains yeast extract that could be an external source of EV-like particles and thus influence

concentration measurement by NTA.

Figure 1a: please include information on how many EV proteins are with or without SP. This might provide further hints towards EV biogenesis in oomycetes.

Figure 1a: must be "protein number" instead of "gene number"

Figure 1d: please, make clear that PsXEG1 is a positive control, and not a EV protein candidate

Figure S1b: in the legend: s.e.m. shown in red, there is nothing red?

Figure 2d: there is a signal of ARA6 visible in the P100 fraction. Was this signal visible in all three replicates?

Figure 3d: please, include the PsTET1 and PsTET3 single ko in the EV analysis to confirm functional redundancy in EV biogenesis, and to clearly correlate to reduced virulence in the tet1tet3 dko.

Please, change y-axis label in accordance to NTA measurement shown in Fig. S1 (concentration in particles/ml).

Figure S5/Figure 3d: how is the growth behavior of tet1tet3 dko in the EV liquid medium? This is crucial to know, since less EV particles are measured in the mutants. How are the particle concentrations normalized for comparison between Ps strains? Please, explain this in the method section, as well.

Figure 3e: better to show the entire CBB gel instead of a random fraction.

Figure 3f: the authors state in the text: These results are consistent with degradation of the external regions of the GFP-PsTET3 fusion, leaving the internally localized GFP domain protected by membrane structures from digestion by trypsin". Shouldn't it be then expected that a size of GFP plus the internal TET3 residue until the first external EC loop appears?

Figure 3g: GFP-TET3 seems to be stronger exposed than the free GFP. This probably makes the entire hypocotyl cell fluorescent in the GFP-TET3 image. How often were these GFP spots observed? More image material and information would be helpful.

Figure S7: please include a Protter (<https://wlab.ethz.ch/protter/start/>) model of the TET3, which provides a nice structural overview of the ECs in TETs.

Figure 4: buffer only is not a good negative control in the experiments for EC2 effects. Please, use another control, for instance EC1.

All TET3 mutants are adequately expressed in tobacco and localized to the plasma membrane (Fig.S4). This indeed supports the idea that TET3 mutants T3M2 and T3M3 failed to induce plant immune response due to failed recognition, but not due to TET3 mis-localization. To further confirm this, please test that T3M2 and T3M3 versions are present in the P100 fraction of Nben EVs as shown for wild type TET3 expressed in planta (Fig.3e).

Figure S9c: what is the interpretation that heat does not deactivate EC2 reaction? Does it mean that EC2 structure does not play any role in immune recognition?

Figure 7a: please, include a second EV internal quantification standard (e.g. total protein conc), in addition to NTA particle concentration. This is in particular important, because tet dko were found to produce less EV particles.

Please include a EV treatment protocol of leaves in the method section.

For EV isolation, a 0.45 μm membrane was used instead of standard 0.22 μm membrane. Since Ps EVs are smaller than 200 nm, better to use a 0.22 μm filter.

Figure 7c: to further proof that PsTET3 in EVs can trigger immune response, EVs from Nben expressing TET3 with native or mutated EC2 can be tested on wild type Nben and Gmax. Another experiment would be to pre-treat PsEVs with trypsin prior leaf application assays. That should avoid any plant immune reaction, if the ECs of TET3 is the major trigger.

Since Ps-EVs trigger immune response, what is its biological purpose for Ps to secrete EVs during infection? What happens with PS infection in Ps-EV pre-treated plants? Does this support and hinder infection?

Line 86: please, change "several" into "three" RxLR.

Please, revise spelling mistakes and typing errors, Latin names in italic, etc. throughout the text.

We thank the editor and reviewers for their thoughtful and detailed input. We have done our best to address all the comments. Our detailed responses are as follows.

Reviewer #1 (Remarks to the Author):

The manuscript NCOMMS-22-34996 by Zhu et al. is an interesting study that seeks to provide a better understanding of the biological roles of extracellular vesicles (EVs) produced by the oomycete pathogen *Phytophthora sojae* during plant infection. This is very challenging work, especially when trying to examine the cargo of the EVs and interaction with the plant's immune system. At this point I find that work, while potentially very exciting and contributing to our understanding of plant pathogen-derived EVs, the data appear too preliminary to be published without additional experiments. Primary concerns expressed were:

Comments: 1. The authors don't use MISEV guidelines although performed the experiments using NTA, TEM and proteomic analysis. Please see <https://www.tandfonline.com/doi/full/10.1080/20013078.2018.1535750>

For example, PsEV isolation/characterization was performed from conditioned media. Precautions such as the presence of particles in the media and percent dead cells should be considered and indicated. This also raises the question whether "buffer" as in several experiments used, was an appropriate control. Furthermore, immunoblot analysis confirming the presence of EV marker proteins should also check for EV depleted proteins. Please also provide experimental details according to MISEV.

Response: Thank you for your suggestions. We use NTA to test the EVs numbers of the medium itself and the P100 fraction of medium. The results showed that there are almost no vesicles in the medium compare to the medium cultured *P. sojae* (Supplementary Fig. 1d). In addition, the culture medium is sterilized at 121 °C for 20 minutes before use. Additionally, we examined *P. sojae* mycelium morphology after 8 days growth in liquid medium to ensure there were no dead cells (Supplementary Fig. 7a, b).

We agree that the buffer was not a good control. In the revised manuscript, we use the P100 fraction from medium without culture of *P. sojae* as a negative control, called "Medium P100". We repeated all the experiments involving "buffer" control and got the similar results. For the "EV depleted proteins", there are fewer studies in the field of plants and *P. sojae* about EVs. We are trying to find the depleted proteins for *P. sojae* EVs. So far, we use "Coomassie Blue Staining" and "ponceau S" to indicate the EV fractions which cannot be dyed.

Comments: 2. The authors don't use sucrose gradients or SEC to improve the purity of the isolated EVs, since P100 includes also larger protein aggregates. Along the same lines, as the TEM images show only individual vesicles, were other structures present in the images that could be contaminants or broken cells?

Response: We carried out the suggested experiments. Sucrose gradients centrifugation was used to further purified the EVs in P100. We could see the TEM images that the size of EVs in P100 was heterogeneous (Supplementary Fig. 1a). After sucrose gradients

centrifugation, the EVs we isolated was in similar size (Supplementary Fig. 1b), and they could still induce plant immunity. Contaminants or broken cells were not observed in low-magnification TEM images provided in Supplementary Fig. 1a. Moreover, sucrose gradients centrifugation was also used to purify the GFP-PsTET3 labeled EVs. We found the EVs would be enriched in fraction 3 with a density of 1.141g/ml (Fig. 3g).

Comments: 3. The authors don't use lipid staining of the EVs, e.g. in combination with NTA or to trace GFP-positive EVs in confocal microscopy. It is also unclear whether the authors use NTA to treat plants with similar EV concentrations. To this end, it would be interesting to observe the results using distinct EVs concentrations.

Response: With lipid staining FM4-64 to treat the GFP-PsTET3 labeled EVs, we could see clear signal in GFP track and RFP track under confocal microscopy and they can completely merge together (Fig. 3i). Plants EVs and *P. sojae* EVs were all tested by NTA. In addition, we measured the protein concentration for each EV sample (Supplementary Fig. 7c and Supplementary Fig. 22c). EV samples would be adjusted to the same particle concentration before use.

We found the ROS burst induced by *P. sojae* EVs in a dose-dependent manner. The lowest concentration for *P. sojae* EVs to induce plant ROS burst is 10^7 particles/ml (Supplementary Fig. 21g).

Comments: 4. Please indicate how many times the experiments were repeated independently with similar findings.

Response: Thanks for your suggestions. We have added the statement in every figure legend.

Comments: Figure 1: How relevant is the identification of 2 RxLR effectors in PsEVs? How does this compare to the overall secretome or proteome of Ps in this culture conditions? Did the authors identify any proteins in PsEVs that would provide insights into their biogenesis/release? Furthermore, given the focus on cell death inducing activities, it is unclear to me why data presented in Figure 7c and 7d were not included here.

Response: We provide other two repeats of MS data, which indicates that these two RxLR effectors are present in all three repeats (Supplementary data table 1). In future studies, it will be necessary to determine if these two effectors are indeed associated with EVs.

We compared the EV proteomic and whole proteomic of *P. sojae* using GO enrichment analysis. The result shows that EVs proteins are mainly enriched in molecular functions, cellular components, and biological processes (Supplementary Fig. 2a).

The last figure shows Figure 7c and 7d comparing EVs from plants and pathogens, because we point out tetraspanins are the key factor in recognition by plants, and plants-derived tetraspanins cannot induce immunity in plant. Additionally, the plants could distinguish between EVs due to their differences in tetraspanins.

Comments: Figure 2: Since the cell death inducing activity appears to depend on BAK1 (correct is NbSERK3a/b), it is unclear to me why PsTET1 and PsTET3 constructs without SP motif were not tested. In 2d it is unclear whether the P100 and S100 were collected

from apoplastic wash fluids. Here, MISEV guidelines should also be considered.

Response: Tetraspanins have four transmembrane regions, but no predicted signal peptide. Although tetraspanins do not have a predicted signal peptide, they may be secreted via untraditional secretory pathway, such as EVs.

Plant EVs were collected from apoplastic wash fluids. We have highlighted this in figure legend and method section.

Comments: Figure 3: It is surprising that the *P. sojae* PsTET1 and PsTET3 double mutants appear to have no general growth defects. Can the authors please comment. 3d, please include e.g. FM4-64 tracing to provide additional evidence of GFP-positive PsEVs at or in plant cells. Also, use isolated GFP-PsEVs and apply to plant cells. How relevant is a Ps control strain expressing cytosolic GFP?

Response: Although tetraspanins are involved in many biological processes, each species has a number of homologs. The roles of different tetraspanins may differ. For example, in *Arabidopsis thaliana*, knocking out AtTET8 does not affect growth but decreased EV secretion¹.

We have added data to figure 3 to demonstrate that the green spots are EVs based on FM4-64. As well, we tested whether plant cells could take up *P. sojae* EVs, but so far no uptake was observed.

Due to a report stating that GFP alone cannot sort EVs, we chose the Ps control strain expressing cytosolic GFP². So we thought that the strain expressing GFP would be a suitable control. The references are listed below.

(1) Liu NJ, Wang N, Bao JJ, Zhu HX, Wang LJ, Chen XY. Lipidomic Analysis Reveals the Importance of GIPCs in Arabidopsis Leaf Extracellular Vesicles. *Molecular plant* 13, 1523-1532 (2020).

(2) Schatz D, Rosenwasser S, Malitsky S, Wolf SG, Feldmesser E, Vardi A. Communication via extracellular vesicles enhances viral infection of a cosmopolitan alga. *Nature microbiology* 2, 1485-1492 (2017).

Comments: Figure 7: Were similar PsEV concentrations applied when collecting EVs from wild type vs mutants? Do the strains produce EVs at similar concentrations? Would proteinase-treated and washed PsEVs still induce cell death?

Response: The NTA results show that PsTET1 and PsTET3 single knockout mutants released same amount of EVs as wild type, but double knockout mutants released fewer EVs (Supplementary Fig. 7c, d and Fig. 3d). Before treating plants, we would adjust the EVs to a same concentration based on the NTA results and protein concentration.

We treated the purified EVs with trypsin, which can degrade EV surface proteins without destroying EV structure (Supplementary Fig. 21a, b). EVs treated and re-purified almost lost their ability to induce ROS bursts in *N. benthamiana* leaves (Supplementary Fig. 21c).

Comments: Figure S4: Can the authors comment what the smaller and larger spots were? Also, GFP-PsTET3 appears to show a signal around the nucleus? And GFP-AtARA6 is plasma membrane while this should be late endosome?

Response: To explain these questions, we co-expressed GFP-PsTET1/GFP-PsTET3 with RFP-AtTET8 (plant EV marker protein), RFP-AtARA6 (plant multivesicular body marker protein) and RFP (has nucleus localization) on *N. benthamiana*¹. The results showed that GFP-PsTET1/GFP-PsTET3 completely colocalized with RFP-AtTET8, which also have the spots like localization, and partially colocalized with RFP-AtARA6 (Fig. 2d). And the results also showed that there is a signal around nucleus (Fig. 2d).

AtARA6 is a marker protein of multivesicular bodies, which is original from late endosome. The reference is list below.

(1) He B, et al. RNA-binding proteins contribute to small RNA loading in plant extracellular vesicles. *Nature plants* 7, 342-352 (2021).

Comments: Figure S5: Please provide data showing the knock-out of PsTET1 and PsTET3 using RT-PCR.

Response: The data showed *PsTET1* or *PsTET3* are not transcript in corresponding knockout mutants and we added it in Supplementary Fig. 5a-c.

Comments: Figure S6: Since GFP-PsTET3 is overexpressed, does it result in more vesicle production and affect virulence?

Response: The NTA results showed that GFP-PsTET3 overexpressed transformants released more EVs compare to wild type, but the virulence significantly decreased (Supplementary Fig. 8d-g). We thought that more EVs released by the strain may induce stronger plant immunity and resistant against the infection.

Comments: Figure S9: Is buffer the best control given expression in yeast and yeast extract stimulating plant immunity?

Response: We agree that buffer is not a suitable control, so we expressed EC1-GFP-His protein in yeast at the same time and repeated the experiments involving “buffer” control (Supplementary Fig. 13a, c). And we got similar results.

Comments: Discussion: It remains unclear why RxLR effectors would be present in EVs, potentially delivered by EVs into plant cells, yet TET proteins present at EVs induce cell death?

Response: As we mentioned above. We detected the two RxLR effectors in all three replicates. In our provide data, PsTET3 is indeed localized on EVs and could induce plant immunity. So far, we cannot exclude the possibility that there are RxLR effectors that can inhibit the plant immunity induced by tetrapanins. This is a very interesting question and we will address it in a future manuscript.

Reviewer #2 (Remarks to the Author):

The authors provide a preliminary report on an EV proteome of *Phytophthora sojae*. They identify candidate tetraspanins (TETs) and report that these trigger cell death when

expressed in *Nicotiana benthamiana* cells, assuming that they are also exported in EVs by the plant host to trigger PTI-associated cell death mediated by BAK1. They characterise the region of TET1 and TET3 that act as PAMPs to activate PTI and show that orthologous regions in plant TET equivalents are divergent and therefore not detected in a similar way. The work is, at best, preliminary and has a number of significant deficiencies that preclude publication.

Major Criticisms:

Comments: 1. The description of EV purification is undetailed. It ultimately consists of a 100,000 xg ultracentrifugation spin to give a pellet (P100). There are no low-magnification images in Sup Fig 1. It is widely recognised that a pellet from a 100k spin will gather a great deal of membrane-associated material. As a result it is a requirement to resuspend and then perform a sucrose gradient to purify EVs.

Response: We carried out the suggested experiments. We purified the EVs following the sucrose gradients centrifugation method for both wild type EVs and GFP-PsTET3 labeled EVs. We found that size of EVs isolate by sucrose gradients centrifugation are more homogeneous (Supplementary Fig. 1b). And GFP-PsTET3 labeled EVs are enriched in Fraction 3 with a density of 1.141 g/ml (Fig. 3g).

Comments: 2. The information on the proteome of the EVs is not acceptable! There are no raw data provided and the protein IDs are not publicly available in accepted international databases. It is thus not possible to cross-check what the proteins are. There is no proteome provided from the supernatant (i.e. conventionally secreted proteins) for comparison. I would expect such comparisons to include volcano plots, for example. There is no comparison of the 'EV' proteome to other EV studies.

Response: Thank you for your suggestions. For the proteome data, we added two replicate data in Supplementary data table 1. In addition, the IDs are available in the website (https://genome.jgi.doe.gov/portal/pages/dynamicOrganismDownload.jsf?organism=Physo1_1), which can be easily accessed. We have to admit that we didn't test the proteins in supernatant after EVs collection. In comparison to the whole proteome of *P. sojae*, we found that EV proteins are mainly enriched in molecular function, cellular component, and biological process (Supplementary Fig. 2a). Additionally, we noticed that *P. capsici* EV proteome and functional analysis of EVs are quite similar to ours¹.

The reference is list below.

(1) Fang Y, Wang Z, Zhang S, Peng Q, Liu X. Characterization and proteome analysis of the extracellular vesicles of *Phytophthora capsici*. *Journal of proteomics* 238, 104137 (2021).

Comments: 3. The authors express candidate EV-associated transmembrane proteins in *N. benthamiana* in Fig 1, assuming they will be exported in plant EVs. They do not provide evidence using electron microscopy or confocal microscopy that these proteins are incorporated in *N. benthamiana* EVs. The evidence that TET proteins are in EVs is provided by a 100k spin (P100) from lysate, rather than apoplast purification (Fig 2d). The proteins could be in any membrane from inside the cell. The evidence in Suppl Fig S4 that

the TET proteins are exported is extremely poor. There is no demonstration that these GFP-labelled proteins are exported using additional markers (e.g. membrane). GFP is degraded in the *N. benthamiana* apoplast. The AtAra6 used as a negative control in Fig 2d is a poor one. Ara6 will also be associated with early endosomes, which would be co-purified in the P100.

Response: So far, we have not yet verified that all the candidate proteins were indeed cargos inside of the EVs. Specifically, we focused on PsTET1 and PsTET3, two cell-death-induced proteins. Our additional data showed these two proteins perfectly colocalized with plant EV maker protein AtTET8 (Fig 2d). It is our regret that we failed to explain in the figure legend that it was the apoplastic wash fluids rather than lysate that were being used to isolate all plant EVs. Lysate fraction was the positive control of the proteins. The corresponding reference articles are listed below. In the plant EV studies, AtTET8 was used as EVs positive marker protein and AtARA6 was used as EV negative proteins, because AtARA6 cannot be released in plant apoplastic fluids although it localized on multivesicular body (MVB)^{1,2,3}.

(1) Rutter BD, Innes RW. Extracellular Vesicles Isolated from the Leaf Apoplast Carry Stress-Response Proteins. *Plant physiology* 173, 728-741 (2017).

(2) Cai Q, et al. Plants send small RNAs in extracellular vesicles to fungal pathogen to silence virulence genes. *Science* 360, 1126-1129 (2018).

(3) He B, et al. RNA-binding proteins contribute to small RNA loading in plant extracellular vesicles. *Nature plants* 7, 342-352 (2021).

Comments: 4. The growth rate is not provided for the double KO lines in Suppl Fig 5 – I cannot see growth over time.

Response: For all knockout mutants, we recorded day-by-day growth rates and found no difference between the mutants and the wild type (Supplementary Fig. 6a, b).

Comments: 5. In Fig 3a 5 TET family members are described. A brief search of the *P. sojae* genome indicates there are 6 family members.

Response: In the genomic of *P. sojae* version 1.1. provided on the website (https://genome.jgi.doe.gov/portal/pages/dynamicOrganismDownload.jsf?organism=Physo1_1). We screened total five genes that code tetraspanin proteins by the method described in method section. The five genes are PsTET1 (ID: Ps_136802, scaffold_48:291834-292745), PsTET2 (ID: Ps_157501, scaffold_48:284989-285951), PsTET3 (ID: Ps_155746, scaffold_7:746485-747552), PsTET4 (ID: Ps_136800, scaffold_48:286869-287777), PsTET5 (ID: Ps_136800, scaffold_48:287935-288843).

Comments: 6. For Fig 3f I would like to see that triton has disrupted the integrity of the EVs, using TEM and nanoparticle tracking, at least. Also, show that trypsin alone does not disrupt the integrity of the EVs.

Response: Thanks for your suggestions. We carried out the suggested experiments. NTA results showed there is no significantly different between trypsin-treated EVs and wild type. However, Triton X-100 can indeed destroy almost all EV particles (Supplementary Fig. 21b).

Comments: 7. In Fig 3g, provide convincing evidence that the green dots are indeed EVs. This is a very preliminary result.

Response: Aside from the green spots observed in infection stage, we isolated the EVs from culture filtrate of GFP-PsTET3 overexpressed strain and observed green fluorescence under confocal microscopy. Moreover, when the GFP-PsTET3 labeled EVs were treated with FM4-64 (lipid staining), red fluorescence can also be observed and merged perfectly with GFP, which strongly indicate that the green spots we observed are EVs (Fig. 3i).

Comments: 8. Expression of TET mutants (Suppl Fig 8) and targeting to the *N. benthamiana* membrane and export into the apoplast (Suppl Fig 4) are not adequately demonstrated in my view. The immunoblots are very poor, as are the confocal images (which have no markers in the plant to be associated with).

Response: Thanks for your suggestions. We agree that is not adequately to say these mutants are still localized on plant EVs. So, we co-expressed PsTET1/PsTET3 with RFP-AtTET8 (plant EV marker protein), RFP-AtARA6 (plant multivesicular body marker protein), then we observed PsTET1/PsTET3 is colocalized with AtTET8 and partially colocalized with AtARA6 (Fig. 2d). Furthermore, the mutants PsTET3M2 and PsTET3M3, which lost the ability to induce cell death can still colocalized with AtTET8 and correctly target to the EVs of *N. benthamiana* (Supplementary Fig. 12a, b).

Comments: 9. In Fig 6, why did they not synthesise the 16aa peptide from oomycetes to show that it triggers PTI cell death, whereas the plant equivalent does not?

Response: In reality, we have a synthetic peptide of 16aa from PsTET3. However, it does not appear to be effective at inducing plant immunity due to unknown factors.

Comments: 10. The BAK1-dependent recognition of TET proteins is based upon expression of TET proteins, or their expression and infiltration, into plant leaves (Fig 4). *P. sojae* EVs also trigger cell death (Fig 7), but do they do so in a BAK1-dependent way?

Response: We carried out the suggested experiments. *P. sojae* EVs induce ROS bursts that are significantly diminished in BAK1-silenced plants (Supplementary Fig. 21e, f).

Minor comments

Comments: Line 172 – reference the PTI marker genes.

Response: The reference is added as follows.

Nie J, Yin Z, Li Z, Wu Y, Huang L. A small cysteine-rich protein from two kingdoms of microbes is recognized as a novel pathogen-associated molecular pattern. *The New phytologist* 222, 995-1011 (2019).

Reviewer #3 (Remarks to the Author):

Jinyi Zhu and colleagues report on extracellular vesicles (EVs), released by the disastrous

soybean pathogen *Phytophthora sojae* (Ps), and the plant recognition of EV-associated Ps tetraspanin (TET) PsTET3 that induces a plant cell death-like immune response in a BAK1-dependnet manner. The role EVs in plant-pathogen interactions is a relative young research field and many novel discoveries are expected in future. Plant TETs have been previously reported as EV biomarkers. Here, a pathogen EV-TET is described for the first time as an immune trigger, therefore extending our knowledge on EV-TETs in plant-pathogen interaction. The manuscript is written well and major findings are displayed in a suitable manner.

Please, find my critical comments to improve the work as follows:

Comments: Figure S1: please, include a NTA measurement of only liquid medium control. This is crucial, because the medium contains yeast extract that could be an external source of EV-like particles and thus influence concentration measurement by NTA.

Response: We agree this information is important. By using NTA, EVs numbers of the medium itself and the P100 fraction of medium were measured. Both showed extremely low concentration (Supplementary Fig. 1d). In addition, the liquid medium is sterilized at 121 °C for 20 minutes before use.

Comments: Figure 1a: please include information on how many EV proteins are with or without SP. This might provide further hinds towards EV biogenesis in oomycetes.

Response: Out of the total 468 EV proteins, 304 lack a predicted signal peptide, according to our analysis (Supplementary Fig. 2b).

Comments: Figure 1a: must be "protein number" instead of "gene number"

Response: Corrected.

Comments: Figure 1d: please, make clear that PsXEG1 is a positive control, and not a EV protein candidate

Response: We added a statement in figure legend of Figure 1 as "PsXEG1 is a cell death induced control rather than the candidate of EV protein".

Comments: Figure S1b: in the legend: s.e.m. shown in red, there is nothing red?

Response: It is corrected as "±s.e.m. shown in dotted line".

Comments: Figure 2d: there is a signal of ARA6 visible in the P100 fraction. Was this signal visible in all three replicates?

Response: It was invisible in other replicates (Supplementary Fig. 12c and Supplementary Fig. 15). This may be caused by lysate leakage near the lane.

Comments: Figure 3d: please, include the PsTET1 and PsTET3 single ko in the EV analysis to confirm functional redundancy in EV biogenesis, and to clearly correlate to reduced virulence in the tet1tet3 dko.

Please, change y-axis label in accordance to NTA measurement shown in Fig. S1

(concentration in particles/ml).

Response: Thank you for your suggestions. We tested the EV concentration of all knockout mutants (single knockout and double knockout) by NTA. Only double knockout mutants showed a decrease in EV concentration, while single knockout mutants did not differ from wild type (Supplementary Fig. 7d and Fig. 3d).

Y-axis label was changed to "concentration in particles/ml" in the revised manuscript.

Comments: Figure S5/Figure 3d: how is the growth behavior of tet1tet3 dko in the EV liquid medium? This is crucial to know, since less EV particles are measured in the mutants. How are the particle concentrations normalized for comparison between Ps strains? Please, explain this in the method section, as well.

Response: We carried out the suggested experiments. When knockout mutants are grown in liquid media, it appears there are no differences in growth behavior and dry weight after 8 days. According to the NTA data, particle concentrations are adjusted to the same level before treating the plants. There is a detailed explanation of this in method section.

Comments: Figure 3e: better to show the entire CBB gel instead of a random fraction.

Response: We showed the entire CBB gel in the revised manuscript.

Comments: Figure 3f: the authors state in the text: "These results are consistent with degradation of the external regions of the GFP-PsTET3 fusion, leaving the internally localized GFP domain protected by membrane structures from digestion by trypsin". Shouldn't it be then expected that a size of GFP plus the internal TET3 residue until the first external EC loop appears?

Response: We agreed that original description was not accurate. It should be a size of GFP plus the internal PsTET3 residue. We have corrected the description. However, the band exist in the gel is accurate because N-terminal of PsTET3 inside the EVs only has 20 amino acids, which are too small to observe a difference in the gel.

Comments: Figure 3g: GFP-TET3 seems to be stronger exposed than the free GFP. This probably makes the entire hypocotyl cell fluorescent in the GFP-TET3 image. How often were these GFP spots observed? More image material and information would be helpful.

Response: All images have been brightened identically to enhance visualization of EVs, which was mentioned in figure legend. We have repeated this experiment at least three times and always observed the green spots near the hyphae of GFP-PsTET3 overexpressed strain rather than free GFP strain. We provide another comparison figures here.

Comments: Figure S7: please include a Protter (<https://wlab.ethz.ch/protter/start/>) model of the TET3, which provides a nice structural overview of the ECs in TETs.

Response: Thank you for your suggestions. We used this website to get a better understanding of tetraspanins structure (Supplementary Fig. 9).

Comments: Figure 4: buffer only is not a good negative control in the experiments for EC2 effects. Please, use another control, for instance EC1.

Response: We agree that “buffer” is not a good negative control. EC1 contains only 25 amino acids, which makes it difficult to purify; therefore, we add a GFP tag and a His tag at its C-terminus. The EC1-GFP-His protein was successfully purified by yeast system along with EC2-His. All experiments involving “buffer” control were repeated using EC1-GFP-His protein as a negative control and the results are similar.

Comments: All TET3 mutants are adequately expressed in tobacco and localized to the plasma membrane (Fig.S4). This indeed supports the idea that TET3 mutants T3M2 and T3M3 failed to induce plant immune response due to failed recognition, but not due to TET3 mis-localization. To further confirm this, please test that T3M2 and T3M3 versions are present in the P100 fraction of Nben EVs as shown for wild type TET3 expressed in planta (Fig.3e).

Response: Thank you for your suggestions. We co-expressed GFP-PsTET3M2/ GFP-PsTET3M3 with RFP-AtTET8 and RFP-AtARA6 on *N. benthamiana*, and we found GFP-PsTET3M2/ GFP-PsTET3M3 colocalized with AtTET8 and partially colocalized with AtARA6 (Supplementary Fig. 12a). In addition, GFP-PsTET3M2 and GFP-PsTET3M3 can still correctly target on plant EVs (Supplementary Fig. 12c).

Comments: Figure S9c: what is the interpretation that heat does not deactivate EC2 reaction? Does it mean that EC2 structure does not play any role in immune recognition?

Response: Although heat-treated dose not effluence the EC2 to induce plant immunity, we cannot conclude conclusively that EC2 structure does not contribute to immune recognition in this case. It should be possible to explain this in the future through structural biology.

Comments: Figure 7a: please, include a second EV internal quantification standard (e.g. total protein conc), in addition to NTA particle concentration. This is in particular important, because tet dko were found to produce less EV particles. Please include a EV treatment protocol of leaves in the method section. For EV isolation, a 0.45 µm membrane was used instead of standard 0.22 µm membrane. Since Ps EVs are smaller than 200 nm, better to use a 0.22 µm filter.

Response: The total protein concentration of different EV samples was tested. The concentration trend is similar to that of NTA (Supplementary Fig. 7c, Supplementary Fig. 8e and Supplementary Fig. 22c).

It is true that the concentration of EVs in wild type and double knockout mutants are different, but in order to treat the plants effectively, we adjusted the concentration to the same level. We added this protocol in method section "Measurement of ROS production".

We used 0.22 µm membrane and got the same results. 0.22 m membrane was used in subsequent experiments. We updated it in method section.

Comments: Figure 7c: to further proof that PsTET3 in EVs can trigger immune response, EVs from Nben expressing TET3 with native or mutated EC2 can be tested on wild type Nben and Gmax. Another experiment would be to pre-treat PsEVs with trypsin prior leaf application assays. That should avoid any plant immune reaction, if the ECs of TET3 is the major trigger.

Response: We expressed PsTET3, PsTET3M3 (still targets plant EV but cannot induce cell death) and NbTET6 on *N. benthamiana*. The EVs were then isolated from apoplastic fluid after two days of expression. Our findings show that only EVs expressing PsTET3 can strongly induce ROS burst on soybean leaves (Fig. 7e, f). Also, we found that *P. sojae* EVs treated with trypsin almost lost the ability to produce ROS burst (Supplementary Fig. 21c, d).

Comments: Since Ps-EVs trigger immune response, what is biological purpose for Ps to secrete EVs during infection? What happens with PS infection in Ps-EV pre-treated plants? Does this support and hinder infection?

Response: We found the ROS burst induced by *P. sojae* EVs showed a dose-dependent manner and when the concentration of EVs is 10^6 particles ml^{-1} or less than this concentration, the ROS burst cannot be induced. So we infiltrate *N. benthamiana* with 10^6 particles ml^{-1} EVs followed inoculated *P. capsici*. The results showed low concentration of EVs would promote the infection of *P. capsica* (Supplementary Fig. 21g-i).

Comments: Line 86: please, change "several" into "three" RxLR.

Response: Corrected.

Comments: Please, revise spelling mistakes and typing errors, Latin names in italic, etc. throughout the text.

Response: We have corrected the writing errors and edited the revised manuscript very carefully.

REVIEWER COMMENTS

Reviewer #1 (Remarks to the Author):

The manuscript has greatly improved, having added requested experiments and controls. I believe it is in principle acceptable for publication, pending substantial english language editing and publication quality figure legends throughout the entire manuscript.

For example, supplementary figure legend 1 states "a is the P100 fraction. b is the fraction after sucrose gradient centrifugation. c is the morphology of single EV". This is insufficient to understand the figure without prior knowledge and reading the main text. Figure legends should be self-explanatory. Please review and correct all figure legends accordingly.

The main text of the manuscript needs substantial language editing. For example, line 87 "Compare to the whole proteomic of *P. sojae*..." Please edit the whole text.

Minor points:

- Supplementary fig. 1d is not cited in the text.
- Please show size profiles of EVs purified over sucrose gradients.
- Which construct was used for BAK1 silencing in *N. benthamiana*?
- Please use the appropriate term for BAK1 in *N. benthamiana*, which is NbSERK3a/b. There are two homologous genes for BAK1 in *N. benthamiana*. See <https://www.ncbi.nlm.nih.gov/pmc/articles/PMC3029390/>
- Which TET sequences are found in symbiotic fungi? Do they have an EC2 domain capable for inducing cell death?
- Supplementary fig. 21 h,i should also include *P. capsici* infection without any pre-treatment.
- Fig. 7e,f show ROS burst elicited by EVs collected from PsTET3 expressing leaves. Please clarify that these leaves do not show cell death?

Reviewer #2 (Remarks to the Author):

My comments are attached

Reviewer #3 (Remarks to the Author):

The authors have made substantial improvements of their manuscript and provide additional information in the text and with new figures. Most of my comments have been seriously addressed I would in principle vote for accepting the manuscript for publication upon minor revision.

I would make the following suggestions to further improve the manuscript:

Figure 3i) overlap GFP and FM4-64 images to really show overlapping signals

include the information (with data) that a synthetic 16 aa peptide from PsTET3 did not trigger plant immunity.

Reviewer #4 (Remarks to the Author):

I have gone through the revised version of the manuscript entitled, "Divergent sequences of tetraspanins enable plants to specifically recognize microbe derived extracellular vesicles" by Zhu et al. All the comments have been addressed efficiently. The proteomic data as well as Real Time expression data have been generated

meticulously and presented very well. Results have been discussed in a meticulous manner.
I will just suggest the authors of check the grammar/ language of manuscript critically. Plz. modify the sentence
line no. 132.
I recommendation acceptance of this manuscript.

Reviewer #2 (Remarks to the Author): The authors provide a preliminary report on an EV proteome of *Phytophthora sojae*. They identify candidate tetraspanins (TETs) and report that these trigger cell death when expressed in *Nicotiana benthamiana* cells, assuming that they are also exported in EVs by the plant host to trigger PTI-associated cell death mediated by BAK1. They characterise the region of TET1 and TET3 that act as PAMPs to activate PTI and show that orthologous regions in plant TET equivalents are divergent and therefore not detected in a similar way. The work is, at best, preliminary and has a number of significant deficiencies that preclude publication.

I have added my additional comments in red below the authors' responses to previous comments.

Major Criticisms:

Comments: 1. The description of EV purification is undetailed. It ultimately consists of a 100,000 xg ultracentrifugation spin to give a pellet (P100). There are no low-magnification images in Sup Fig 1. It is widely recognised that a pellet from a 100k spin will gather a great deal of membrane-associated material. As a result it is a requirement to resuspend and then perform a sucrose gradient to purify EVs.

Response: We carried out the suggested experiments. We purified the EVs following the sucrose gradients centrifugation method for both wild type EVs and GFP-PsTET3 labeled EVs. We found that size of EVs isolate by sucrose gradients centrifugation are more homogeneous (Supplementary Fig. 1b). And GFP-PsTET3 labeled EVs are enriched in Fraction 3 with a density of 1.141 g/ml (Fig. 3g).

Further comment: the authors appear to have done very little to meet criticisms relating to EV isolation regarding Suppl Fig 1. They claim that they see more homogeneous EVs from sucrose gradients in Supplementary Figure 1b. However, the NTA (Suppl Fig 1d) is the same as the previous submission, which is based upon P100 rather than sucrose gradient, and they have simply added the media results from an updated P100. The media result is supposed to be a control for the EV samples and should be performed coincidentally. Where is the NTA from sucrose gradients?

Comments: 2. The information on the proteome of the EVs is not acceptable! There are no raw data provided and the protein IDs are not publicly available in accepted international databases. It is thus not possible to cross-check what the proteins are. There is no proteome provided from the supernatant (i.e. conventionally secreted proteins) for comparison. I would expect such comparisons to include volcano plots, for example. There is no comparison of the 'EV' proteome to other EV studies.

Response: Thank you for your suggestions. For the proteome data, we added two replicate data in Supplementary data table 1. In addition, the IDs are available in the website (https://genome.jgi.doe.gov/portal/pages/dynamicOrganismDownload.jsf?organism=Physo1_1), which can be easily accessed.

Further comment: the *Phytophthora sojae* genome is available on public repositories such as Ensembl_Genomes, NCBI, Genbank, FungiDB, UniProt, JGI. When I put in the IDs given in Supplementary Table 1 they are not recognised. It is very important that people can easily access and compare gene/protein IDs. I urge you to convert the IDs you are using to those that are publicly available. I went to the JGI website indicated and tried to access version 1.1. This has been replaced by version 3. Inputting the IDs used in this manuscript gave no matches.

We have to admit that we didn't test the proteins in supernatant after EVs collection. In comparison to the whole proteome of *P. sojae*, we found that EV proteins are mainly enriched in molecular function, cellular component, and biological process (Supplementary Fig. 2a). Additionally, we

noticed that *P. capsici* EV proteome and functional analysis of EVs are quite similar to ours¹. The reference is list below. (1) Fang Y, Wang Z, Zhang S, Peng Q, Liu X. Characterization and proteome analysis of the extracellular vesicles of *Phytophthora capsici*. *Journal of proteomics* 238, 104137 (2021).

Further comment: It is difficult to comment on the EV proteome presented here. There is no comparison to the proteome from the supernatant after the P100 spin, and I could not find evidence of comparisons made to the *P. capsici* EV proteome, which the authors indicate above is quite similar.

Comments: 3. The authors express candidate EV-associated transmembrane proteins in *N. benthamiana* in Fig 1, assuming they will be exported in plant EVs. They do not provide evidence using electron microscopy or confocal microscopy that these proteins are incorporated in *N. benthamiana* EVs. The evidence that TET proteins are in EVs is provided by a 100k spin (P100) from lysate, rather than apoplast purification (Fig 2d). The proteins could be in any membrane from inside the cell. The evidence in Suppl Fig S4 that the TET proteins are exported is extremely poor. There is no demonstration that these GFP-labelled proteins are exported using additional markers (e.g. membrane). GFP is degraded in the *N. benthamiana* apoplast. The AtAra6 used as a negative control in Fig 2d is a poor one. Ara6 will also be associated with early endosomes, which would be co-purified in the P100.

Response: So far, we have not yet verified that all the candidate proteins were indeed cargos inside of the EVs. Specifically, we focused on PsTET1 and PsTET3, two cell-deathinduced proteins. Our additional data showed these two proteins perfectly colocalized with plant EV maker protein AtTET8 (Fig 2d). It is our regret that we failed to explain in the figure legend that it was the apoplastic wash fluids rather than lysate that were being used to isolate all plant EVs. Lysate fraction was the positive control of the proteins. The corresponding reference articles are listed below. In the plant EV studies, AtTET8 was used as EVs positive marker protein and AtARA6 was used as EV negative proteins, because AtARA6 cannot be released in plant apoplastic fluids although it localized on multivesicular body (MVB)^{1,2,3}.

(1) Rutter BD, Innes RW. Extracellular Vesicles Isolated from the Leaf Apoplast Carry Stress-Response Proteins. *Plant physiology* 173, 728-741 (2017).

(2) Cai Q, et al. Plants send small RNAs in extracellular vesicles to fungal pathogen to silence virulence genes. *Science* 360, 1126-1129 (2018).

(3) He B, et al. RNA-binding proteins contribute to small RNA loading in plant extracellular vesicles. *Nature plants* 7, 342-352 (2021).

Further comment: the new confocal images added to Fig 2d are not convincing at all. The GFP-PsTET1 and 3 co-expressed with free RFP show green fluorescence around the nucleus, indicative of ER-association. Co-expression with RFP-AtTET8 in each case looks as if the cells are very unwell (the images are too poor to make out details, but both proteins also appear to be in nuclei). Moreover, the AtAra6 localisation does not look as if the marker protein is associated with MVBs – more then that it appears to largely co-localise with GFP-PsTET proteins. I cannot get much of value from these images, and I can't see whether there are EVs outside of plant cells associated with PsTET proteins. It would make more sense to co-express GFP-PsTET1 and 3 with RFP-AtTET8 and to purify EVs from AWF by P100 and glucose gradient and then look at co-localisation to EVs under confocal, as has been done previously by the Jin group.

Comments: 4. The growth rate is not provided for the double KO lines in Suppl Fig 5 – I cannot see growth over time.

Response: For all knockout mutants, we recorded day-by-day growth rates and found no difference between the mutants and the wild type (Supplementary Fig. 6a, b).

Further comment: the authors have addressed this point, although I would have expected graphs of growth curves over time with statistical analyses from different replicates, rather than selected pictures.

Comments: 5. In Fig 3a 5 TET family members are described. A brief search of the *P. sojae* genome indicates there are 6 family members.

Response: In the genomic of *P. sojae* version 1.1. provided on the website (https://genome.jgi.doe.gov/portal/pages/dynamicOrganismDownload.jsf?organism=Physo1_1). We screened total five genes that code tetraspanin proteins by the method described in method section. The five genes are PsTET1 (ID: Ps_136802, scaffold_48:291834-292745), PsTET2 (ID: Ps_157501, scaffold_48:284989-285951), PsTET3 (ID: Ps_155746, scaffold_7:746485-747552), PsTET4 (ID: Ps_136800, scaffold_48:286869-287777), PsTET5 (ID: Ps_136800, scaffold_48:287935-288843).

Further comment: Again, the IDs given above are not available on public databases containing the *P. sojae* genome. I went to the JGI website indicated and tried to access version 1.1. This has been replaced by version 3. Inputting the IDs used in this manuscript gave no matches. Please ensure that your sequences are accessible.

Comments: 6. For Fig 3f I would like to see that triton has disrupted the integrity of the EVs, using TEM and nanoparticle tracking, at least. Also, show that trypsin alone does not disrupt the integrity of the EVs.

Response: Thanks for your suggestions. We carried out the suggested experiments. NTA results showed there is no significantly different between trypsin-treated EVs and wild type. However, Triton X-100 can indeed destroy almost all EV particles (Supplementary Fig. 21b).

Further comment: The authors have addressed this. However, it would be good to know whether they expect the GFP-PsTET3 to be detectable following triton treatment of EVs in Fig 3f?

Comments: 7. In Fig 3g, provide convincing evidence that the green dots are indeed EVs. This is a very preliminary result.

Response: Aside from the green spots observed in infection stage, we isolated the EVs from culture filtrate of GFP-PsTET3 overexpressed strain and observed green fluorescence under confocal microscopy. Moreover, when the GFP-PsTET3 labeled EVs were treated with FM4-64 (lipid staining), red fluorescence can also be observed and merged perfectly with GFP, which strongly indicate that the green spots we observed are EVs (Fig. 3i).

Further comment: Addressed - The use of the FM4-64 helps to confirm that the GFP-PsTET3 dots are membranous, if not proving they are EVs.

Comments: 8. Expression of TET mutants (Suppl Fig 8) and targeting to the *N. benthamiana* membrane and export into the apoplast (Suppl Fig 4) are not adequately demonstrated in my view. The immunoblots are very poor, as are the confocal images (which have no markers in the plant to be associated with).

Response: Thanks for your suggestions. We agree that is not adequately to say these mutants are still localized on plant EVs. So, we co-expressed PsTET1/PsTET3 with RFP-AtTET8 (plant EV marker protein), RFP-AtARA6 (plant multivesicular body marker protein), then we observed PsTET1/PsTET3 is colocalized with AtTET8 and partially colocalized with AtARA6 (Fig. 2d). Furthermore, the mutants PsTET3M2 and PsTET3M3, which lost the ability to induce cell death can still colocalized with AtTET8 and correctly target to the EVs of *N. benthamiana* (Supplementary Fig. 12a, b).

Further comment: I refer to my earlier comment about Fig 2d (and add the Suppl Figure 12a, b) – the images are really not good enough to see anything of any value. The upper images with AtTET8 look very unhealthy.

Comments: 9. In Fig 6, why did they not synthesise the 16aa peptide from oomycetes to show that it triggers PTI cell death, whereas the plant equivalent does not?

Response: In reality, we have a synthetic peptide of 16aa from PsTET3. However, it does not appear to be effective at inducing plant immunity due to unknown factors.

Further comment: Do the authors have an explanation for that? It seems to be pretty crucial in the context of identifying and defining a PAMP.

Comments: 10. The BAK1-dependent recognition of TET proteins is based upon expression of TET proteins, or their expression and infiltration, into plant leaves (Fig 4). *P. sojae* EVs also trigger cell death (Fig 7), but do they do so in a BAK1-dependent way?

Response: We carried out the suggested experiments. *P. sojae* EVs induce ROS bursts that are significantly diminished in BAK1-silenced plants (Supplementary Fig. 21e, f).

Further comment: Addressed

Minor comments Comments: Line 172 – reference the PTI marker genes.

Response: The reference is added as follows. Nie J, Yin Z, Li Z, Wu Y, Huang L. A small cysteine-rich protein from two kingdoms of microbes is recognized as a novel pathogen-associated molecular pattern. *The New phytologist* 222, 995-1011 (2019).

Further comment: please go through all revised texts to ensure the English is grammatically correct.

We thank the editor and reviewers for their thoughtful and detailed input. We have done our best to address all the comments. Our detailed responses (indicated as “Further response”) are as follows.

Reviewer #1 (Remarks to the Author):

The manuscript has greatly improved, having added requested experiments and controls. I believe it is in principle acceptable for publication, pending substantial english language editing and publication quality figure legends throughout the entire manuscript.

For example, supplementary figure legend 1 states "a is the P100 fraction. b is the fraction after sucrose gradient centrifugation. c is the morphology of single EV". This is insufficient to understand the figure without prior knowledge and reading the main text. Figure legends should be self-explanatory. Please review and correct all figure legends accordingly.

Further response: Thanks for your suggestions. The entire text has been revised by a native English speaker.

The main text of the manuscript needs substantial language editing. For example, line 87 "Compare to the whole proteomic of *P. sojae*..." Please edit the whole text.

Further response: Thanks for your suggestions. The entire text has been revised by a native English speaker.

Minor points:

- Supplementary fig. 1d is not cited in the text.

Further response: Added in the text line 80.

- Please show size profiles of EVs purified over sucrose gradients.

Further response: We measured the size of EVs purified by sucrose gradient and found that the EVs exhibited greater homogeneous, with the main peak observed at 112 nm, which is consistent with the TEM results (Supplementary fig. 1d).

- Which construct was used for BAK1 silencing in *N. benthamiana*?

Further response: TRV2 vector was used to silence the NbBAK1/NbSERK3a/b genes in our experiments 1.

(1) Senthil-Kumar M, Mysore KS. Tobacco rattle virus-based virus-induced gene silencing in *Nicotiana benthamiana*. *Nat Protoc* 9, 1549-1562 (2014)

- Please use the appropriate term for BAK1 in *N. benthamiana*, which is NbSERK3a/b. There are two homologous genes for BAK1 in *N. benthamiana*. See <https://www.ncbi.nlm.nih.gov/pmc/articles/PMC3029390/>

Further response: Thanks for your suggestions. We have changed the description about NbSERK3a/b throughout the text.

- Which TET sequences are found in symbiotic fungi? Do they have an EC2 domain capable for inducing cell death?

Further response: The question you raised about symbiotic fungi is very interesting. And we will do it in the next work.

- Supplementary fig. 21 h,i should also include *P. capsici* infection without any pre-treatment.

Further response: We repeated this experiment with a control group that received no treatment, and obtained similar result. The result is exhibited in Supplementary Fig. 22.

- Fig. 7e,f show ROS burst elicited by EVs collected from PsTET3 expressing leaves. Please clarify that these leaves do not show cell death?

Further response: We have clarified that the EVs were isolated from plant leaves before cell death occurred in all relevant experiments. And we presented the condition of the leaves at different time point after agro-infiltration (Supplementary Fig. 5b).

Reviewer #2 (Remarks to the Author):

Reviewer #2 (Remarks to the Author): The authors provide a preliminary report on an EV proteome of *Phytophthora sojae*. They identify candidate tetraspanins (TETs) and report that these trigger cell death when expressed in *Nicotiana benthamiana* cells, assuming that they are also exported in EVs by the plant host to trigger PTI-associated cell death mediated by BAK1. They characterise the region of TET1 and TET3 that act as PAMPs to activate PTI and show that orthologous regions in plant TET equivalents are divergent and therefore not detected in a similar way. The work is, at best, preliminary and has a number of significant deficiencies that preclude publication.

I have added my additional comments in red below the authors' responses to previous comments.

Major Criticisms:

Comments: 1. The description of EV purification is undetailed. It ultimately consists of a 100,000 xg ultracentrifugation spin to give a pellet (P100). There are no low-magnification images in Sup Fig 1. It is widely recognised that a pellet from a 100k spin will gather a great deal of membrane-associated material. As a result it is a requirement to resuspend and then perform a sucrose gradient to purify EVs.

Response: We carried out the suggested experiments. We purified the EVs following the sucrose gradients centrifugation method for both wild type EVs and GFP-PsTET3 labeled EVs. We found that size of EVs isolate by sucrose gradients centrifugation are more homogeneous (Supplementary Fig.1b). And GFP-PsTET3 labeled EVs are enriched in Fraction 3 with a density of 1.141 g/ml (Fig. 3g).

Further comment: the authors appear to have done very little to meet criticisms relating to EV isolation regarding Suppl Fig 1. They claim that they see more homogeneous EVs

from sucrose gradients in Supplementary Figure 1b. However, the NTA (Suppl Fig 1d) is the same as the previous submission, which is based upon P100 rather than sucrose gradient, and they have simply added the media results from an updated P100. The media result is supposed to be a control for the EV samples and should be performed coincidentally. Where is the NTA from sucrose gradients?

Further response: We measured the size of EVs purified by sucrose gradient and found that the EVs exhibited more homogeneous, with the main peak observed at 112 nm, which is consistent with the TEM results (Supplementary fig. 1d).

Comments: 2. The information on the proteome of the EVs is not acceptable! There are no raw data provided and the protein IDs are not publicly available in accepted international databases. It is thus not possible to cross-check what the proteins are. There is no proteome provided from the supernatant (i.e. conventionally secreted proteins) for comparison. I would expect such comparisons to include volcano plots, for example. There is no comparison of the 'EV' proteome to other EV studies.

Response: Thank you for your suggestions. For the proteome data, we added two replicate data in Supplementary data table 1. In addition, the IDs are available in the website (https://genome.jgi.doe.gov/portal/pages/dynamicOrganismDownload.jsf?organism=Physo1_1), which can be easily accessed.

Further comment: the *Phytophthora sojae* genome is available on public repositories such as Ensembl_Genomes, NCBI, Genbank, FungiDB, UniProt, JGI. When I put in the IDs given in Supplementary Table 1 they are not recognised. It is very important that people can easily access and compare gene/protein IDs. I urge you to convert the IDs you are using to those that are publicly available. I went to the JGI website indicated and tried to access version 1.1. This has been replaced by version 3. Inputting the IDs used in this manuscript gave no matches.

Further response: We added the gene IDs of version 3.0 in Supplementary Table 1 with version 1.1. However, when we tried to change all IDs to version 3.0, we cannot find all corresponding IDs in version 3.0 because of the annotation. For example, Although Ps_136802 (PsTET1) cannot be found in version 3.0, we cloned *PstTET1* successfully from cDNA of *P. sojae* and detected its expression level was increased during infection (Fig. 3a and supplementary Fig. 6c). And we checked the data base in JGI website, indeed it appeared version 1.1 was replaced by version 3.0. But version 1.1 still can be downloaded in JGI website. Additionally, we submitted supplements about version 1.1 with this revision.

Response: We have to admit that we didn't test the proteins in supernatant after EVs collection. In comparison to the whole proteome of *P. sojae*, we found that EV proteins are mainly enriched in molecular function, cellular component, and biological process (Supplementary Fig. 2a). Additionally, we noticed that *P. capsici* EV proteome and functional analysis of EVs are quite similar to ours¹. The reference is list below.

(1) Fang Y, Wang Z, Zhang S, Peng Q, Liu X. Characterization and proteome analysis of the extracellular vesicles of *Phytophthora capsici*. Journal of proteomics 238, 104137

(2021).

Further comment: It is difficult to comment on the EV proteome presented here. There is no comparison to the proteome from the supernatant after the P100 spin, and I could not find evidence of comparisons made to the *P. capsici* EV proteome, which the authors indicate above is quite similar.

Further response: We attempted to compare the EV proteome of *P. capsici* with our EV proteome from *P. sojae*. However, the published paper did not provide the complete *P. capsici* EV proteome. According to the text, their gene ontology (GO) analysis showed the proteins are primarily involved in 1) translation, 2) protein, carbohydrate, lipase, and phosphorous metabolism, 3) oxidation/reduction, and 4) transport. In our analysis, the *P. sojae* EV proteins also showed these functions. And in *P. sojae* EV proteome, we identified a total of 468 proteins, out of which 304 proteins lacked a signal peptide, which is similar with previously publications (Supplementary Fig. 2a, b).

Comments: 3. The authors express candidate EV-associated transmembrane proteins in *N. benthamiana* in Fig 1, assuming they will be exported in plant EVs. They do not provide evidence using electron microscopy or confocal microscopy that these proteins are incorporated in *N. benthamiana* EVs. The evidence that TET proteins are in EVs is provided by a 100k spin (P100) from lysate, rather than apoplast purification (Fig 2d). The proteins could be in any membrane from inside the cell. The evidence in Suppl Fig S4 that the TET proteins are exported is extremely poor. There is no demonstration that these GFP-labelled proteins are exported using additional markers (e.g. membrane). GFP is degraded in the *N. benthamiana* apoplast. The AtAra6 used as a negative control in Fig 2d is a poor one. Ara6 will also be associated with early endosomes, which would be copurified in the P100.

Response: So far, we have not yet verified that all the candidate proteins were indeed cargos inside of the EVs. Specifically, we focused on PsTET1 and PsTET3, two cell-death induced proteins. Our additional data showed these two proteins perfectly colocalized with plant EV maker protein AtTET8 (Fig 2d). It is our regret that we failed to explain in the figure legend that it was the apoplastic wash fluids rather than lysate that were being used to isolate all plant EVs. Lysate fraction was the positive control of the proteins. The corresponding reference articles are listed below. In the plant EV studies, AtTET8 was used as EVs positive marker protein and AtARA6 was used as EV negative proteins, because AtARA6 cannot be released in plant apoplastic fluids although it localized on multivesicular body (MVB)^{1,2,3}.

(1) Rutter BD, Innes RW. Extracellular Vesicles Isolated from the Leaf Apoplast Carry Stress-Response

Proteins. *Plant physiology* 173, 728-741 (2017).

(2) Cai Q, et al. Plants send small RNAs in extracellular vesicles to fungal pathogen to silence

virulence genes. *Science* 360, 1126-1129 (2018).

(3) He B, et al. RNA-binding proteins contribute to small RNA loading in plant extracellular vesicles. *Nature plants* 7, 342-352 (2021).

Further comment: the new confocal images added to Fig 2d are not convincing at all. The GFP-PsTET1 and 3 co-expressed with free RFP show green fluorescence around the nucleus, indicative of ER-association. Co-expression with RFP-AtTET8 in each case looks as if the cells are very unwell (the images are too poor to make out details, but both proteins also appear to be in nuclei). Moreover, the AtAra6 localisation does not look as if the marker protein is associated with MVBs – more than that it appears to largely co-localise with GFP-PsTET proteins. I cannot get much of value from these images, and I can't see whether there are EVs outside of plant cells associated with PsTET proteins. It would make more sense to co-express GFP-PsTET1 and 3 with RFP-AtTET8 and to purify EVs from AWF by P100 and glucose gradient and then look at co-localisation to EVs under confocal, as has been done previously by the Jin group.

Further response: Thanks for your suggestions. We co-expressed GFP-PsTET1 and GFP-PsTET3 along with RFP-AtTET8 and purified EVs from AWF by P100 and glucose gradient. We then examined the co-localization of these proteins with the EVs marker AtTET8 under confocal microscopy. In isolated EVs, we found PsTET1 and PsTET3 proteins colocalized with AtTET8 (Supplementary Fig. 5a). Additionally, we presented the condition of leaves at different time point after agro-infiltration (Supplementary Fig. 5b). EVs isolation and leaf collection for confocal microscopy were conducted before 48 hours post infiltration (cell death appeared after 72 hours post infiltration) (Supplementary Fig. 5b). We clarified that there was no observable difference in leaves between infiltrated and untreated simples during the confocal experiments and EVs isolation.

Comments: 4. The growth rate is not provided for the double KO lines in Suppl Fig 5 – I cannot see growth over time.

Response: For all knockout mutants, we recorded day-by-day growth rates and found no difference between the mutants and the wild type (Supplementary Fig. 6a, b).

Further comment: the authors have addressed this point, although I would have expected graphs of growth curves over time with statistical analyses from different replicates, rather than selected pictures.

Further response: We changed that graph to growth curve (Supplementary Fig. 7b).

Comments: 5. In Fig 3a 5 TET family members are described. A brief search of the *P. sojae* genome indicates there are 6 family members.

Response: In the genomic of *P. sojae* version 1.1. provided on the website (https://genome.jgi.doe.gov/portal/pages/dynamicOrganismDownload.jsf?organism=Physo1_1). We screened total five genes that code tetraspanin proteins by the method described in method section. The five genes are PsTET1 (ID: Ps_136802, scaffold_48:291834-292745), PsTET2 (ID: Ps_157501, scaffold_48:284989-285951), PsTET3 (ID: Ps_155746, scaffold_7:746485-747552), PsTET4 (ID: Ps_136800, scaffold_48:286869-287777), PsTET5 (ID: Ps_136800, scaffold_48:287935-288843).

Further comment: Again, the IDs given above are not available on public databases containing the *P. sojae* genome. I went to the JGI website indicated and tried to access version 1.1. This has been replaced by version 3. Inputting the IDs used in this manuscript gave no matches. Please ensure that your sequences are accessible.

Further response: Please see the further response to the comment 2.

Comments: 6. For Fig 3f I would like to see that triton has disrupted the integrity of the EVs, using TEM and nanoparticle tracking, at least. Also, show that trypsin alone does not disrupt the integrity of the EVs.

Response: Thanks for your suggestions. We carried out the suggested experiments. NTA results showed there is no significantly different between trypsin-treated EVs and wild type. However, Triton X-100 can indeed destroy almost all EV particles (Supplementary Fig. 21b).

Further comment: The authors have addressed this. However, it would be good to know whether they expect the GFP-PsTET3 to be detectable following triton treatment of EVs in Fig 3f?

Further response: As it showed in Fig 3f, lane 2, GFP-PsTET3 can be detected after treatment of EVs with Triton x100. A similar experiment has also been performed in previously publications ^{1,2}.

(1) Rutter BD, Innes RW. Extracellular Vesicles Isolated from the Leaf Apoplast Carry Stress-Response Proteins. *Plant physiology* 173, 728-741 (2017)

(2) He B, et al. RNA-binding proteins contribute to small RNA loading in plant extracellular vesicles. *Nature plants* 7, 342-352 (2021)

Comments: 7. In Fig 3g, provide convincing evidence that the green dots are indeed EVs. This is a very preliminary result.

Response: Aside from the green spots observed in infection stage, we isolated the EVs from culture filtrate of GFP-PsTET3 overexpressed strain and observed green fluorescence under confocal microscopy. Moreover, when the GFP-PsTET3 labeled EVs were treated with FM4-64 (lipid staining), red fluorescence can also be observed and merged perfectly with GFP, which strongly indicate that the green spots we observed are EVs (Fig. 3i).

Further comment: Addressed - The use of the FM4-64 helps to confirm that the GFP-PsTET3 dots are membranous, if not proving they are EVs.

Further response: Thanks.

Comments: 8. Expression of TET mutants (Suppl Fig 8) and targeting to the *N. benthamiana* membrane and export into the apoplast (Suppl Fig 4) are not adequately demonstrated in my view. The immunoblots are very poor, as are the confocal images (which have no markers in the plant to be associated with).

Response: Thanks for your suggestions. We agree that is not adequately to say these mutants are still localized on plant EVs. So, we co-expressed PsTET1/PsTET3 with RFP-AtTET8 (plant EV marker protein), RFP-AtARA6 (plant multivesicular body marker protein), then we observed PsTET1/PsTET3 is colocalized with AtTET8 and partially colocalized with AtARA6 (Fig. 2d). Furthermore, the mutants PsTET3M2 and PsTET3M3, which lost the ability to induce cell death can still colocalized with AtTET8 and correctly target to the EVs of *N. benthamiana* (Supplementary Fig. 12a, b).

Further comment: I refer to my earlier comment about Fig 2d (and add the Suppl Figure

12a, b) –the images are really not good enough to see anything of any value. The upper images with AtTET8 look very unhealthy.

Further response: Please see the further response to the comment 3.

Comments: 9. In Fig 6, why did they not synthesise the 16aa peptide from oomycetes to show that it triggers PTI cell death, whereas the plant equivalent does not?

Response: In reality, we have a synthetic peptide of 16aa from PsTET3. However, it does not appear to be effective at inducing plant immunity due to unknown factors.

Further comment: Do the authors have an explanation for that? It seems to be pretty crucial in the context of identifying and defining a PAMP.

Further response: We have made efforts to answer why the synthetic peptide of 16 amino acids did not exhibit any activating activity. However, to date, we have not obtained any experimental results. We will keep to solve this question in future.

We have predicted the stability of this peptide segment, and the results indicate that it has relatively lower stability compared to flg22 and elf18. The predicted website is <https://web.expasy.org/protparam/>. And the predicted information is as follows:

flg22: QRLSTGSRINSAKDDAAGLQIA

The instability index (II) is computed to be 29.26

This classifies the protein as stable.

Aliphatic index: 89.09

Grand average of hydropathicity (GRAVY): -0.477

elf18: SKEKFERTKP HVNVGTIG

The instability index (II) is computed to be -4.18

This classifies the protein as stable.

Aliphatic index: 53.89

Grand average of hydropathicity (GRAVY): -1.044

PsTET3-16aa: ECRSVFYELVEKWTNV

The instability index (II) is computed to be 61.17

This classifies the protein as unstable.

Aliphatic index: 78.75

Grand average of hydropathicity (GRAVY): -0.275

Comments: 10. The BAK1-dependent recognition of TET proteins is based upon expression of TET proteins, or their expression and infiltration, into plant leaves (Fig 4). P. sojae EVs also trigger cell death (Fig 7), but do they do so in a BAK1-dependent way?

Response: We carried out the suggested experiments. P. sojae EVs induce ROS bursts that are significantly diminished in BAK1-silenced plants (Supplementary Fig. 21e, f).

Further comment: Addressed

Further response: Thanks.

Minor comments Comments: Line 172 – reference the PTI marker genes.

Response: The reference is added as follows. Nie J, Yin Z, Li Z, Wu Y, Huang L. A small

cysteine-rich protein from two kingdoms of microbes is recognized as a novel pathogen-associated molecular pattern. *The New phytologist* 222, 995-1011 (2019).

Further comment: please go through all revised texts to ensure the English is grammatically correct.

Further response: Thanks for your suggestions. The entire text has been revised by a native English speaker.

Reviewer #3 (Remarks to the Author):

The authors have made substantial improvements of their manuscript and provide additional information in the text and with new figures. Most of my comments have been seriously addressed I would in principle vote for accepting the manuscript for publication upon minor revision.

I would make the following suggestions to further improve the manuscript:

Figure 3i) overlap GFP and FM4-64 images to really show overlapping signals

Further response: We found the signal was overlapped and showed the relative fluorescence signal curve (Fig. 3i).

include the information (with data) that a synthetic 16 aa peptide from PsTET3 did not trigger plant immunity.

Further response: We have added the related data (Supplementary Fig. 15c).

Reviewer #4 (Remarks to the Author):

I have gone through the revised version of the manuscript entitled, "Divergent sequences of tetraspanins enable plants to specifically recognize microbe derived extracellular vesicles" by Zhu et al. All the comments have been addressed efficiently. The proteomic data as well as Real Time expression data have been generated meticulously and presented very well. Results have been discussed in a meticulous manner.

I will just suggest the authors of check the grammar/ language of manuscript critically. Plz. modify the sentence line no. 132.

I recommend acceptance of this manuscript.

Further response: Thanks for your suggestions. The entire text has been revised by a native English speaker.

REVIEWERS' COMMENTS

Reviewer #2 (Remarks to the Author):

The authors have made a number of additions that have improved the manuscript and have tried to address all comments. One or two sequences are missing in the new annotation 3.0. But they indicate that v1.1 may still be downloadable. They could also independently submit the PsTET sequences to databases.

The confocal images in Fig 2 are, as I said previously, not of good quality. The authors have added images of EVs in Suppl Fig 5 that help to show colocalisation of PsTET and AtTET in vesicle-like structures.

We thank the editor and reviewers for their thoughtful and detailed input. Our detailed responses are as follows.

Reviewer #2 (Remarks to the Author):

The authors have made a number of additions that have improved the manuscript and have tried to address all comments. One or two sequences are missing in the new annotation 3.0. But they indicate that v1.1 may still be downloadable. They could also independently submit the PsTET sequences to databases.

Response: We added sequence information of the 5 tetraspanins of *P. sojae* in Supplementary Table 5 (Cited in main text line 253) as a dataset, which include gene IDs, gene names, amino acid sequences, Nucleotide sequences (CDS) and coordinate.

The confocal images in Fig 2 are, as I said previously, not of good quality. The authors have added images of EVs in Suppl Fig 5 that help to show colocalisation of PsTET and AtTET in vesicle-like structures.

Response: Thanks.